# Loss of vegetation functions during the Paleocene–Eocene Thermal Maximum

Julian Rogger [1,2] ✉, Vera A. Korasidis [3], Gabriel J. Bowen [4], Christine A. Shields [5], Taras V. Gerya[1] & Loïc Pellissier [2,6]

The Paleocene–Eocene Thermal Maximum (PETM) around 56 million years ago was a 5–6°C global warming event that lasted for approximately 200 kyr. A warming-induced loss and a 70–100 kyr lagged recovery of biospheric carbon stocks was suggested to have contributed to the long duration of the climate perturbation. Here, we use a trait-based, eco-evolutionary vegetation model to test whether the PETM warming exceeded the adaptation capacity of vegetation systems, impacting the efficiency of terrestrial organic carbon sequestration and silicate weathering. Combined model simulations and vegetation reconstructions using PETM palynofloras suggest that warming-induced migration and evolutionary adaptation of vegetation were insufficient to prevent a widespread loss of productivity. We conclude that global warming of the magnitude as during the PETM could exceed the response capacity of vegetation systems and cause a long-lasting decline in the efficiency of vegetation-mediated climate regulation mechanisms.

During the onset of the Paleocene–Eocene Thermal Maximum (PETM), a geologically abrupt release of several thousand petagrams of isotopically light carbon, likely of mixed methanogenic, organic and volcanic origin[1], increased atmospheric $CO_2$ by more than 1000 ppm and global average temperatures by 5–6 °C[1,2]. With the estimated mass of carbon release and the magnitude of warming being comparable to future anthropogenic emission scenarios[1], the PETM represents one of the most important geologic analogues to present-day climate change and is key for understanding carbon cycling and biological dynamics in response to a period of severe global warming[1,3].

Paleobotanical records from several locations around the globe suggest that the PETM induced a major disruption and reorganisation of terrestrial vegetation systems through plant range shifts and changes in species composition[4–8], potentially impacting their climate-regulating functions[9]. Geochemical proxies of the atmosphere-ocean $\delta^{13}C$ signature during the period (Fig. 1A, B) indicate that changes in vegetation-mediated organic carbon cycling and storage may have strongly shaped the PETM carbon dynamics. Similar to other large carbon reservoirs, such as methane hydrates[10] or permafrost carbon[11], organic carbon stored in vegetation and soils may have acted as a carbon cycle capacitor[9,12]: contributing to a global negative $\delta^{13}C$ isotope excursion with the release of isotopically light carbon during the onset of the hyperthermal, remaining in a state of reduced carbon storage during an approximately 70–100 kyr period of low $\delta^{13}C$ (the PETM body), and contributing to the carbon cycle and climate recovery as vegetation and soil carbon stocks regrew during the final stages of the event (the PETM recovery). High rates of $\delta^{13}C$ recovery during the termination of the PETM carbon isotope excursion support that a recovery of the photosynthetic sequestration of isotopically light carbon may have contributed to the termination of the PETM[9,12,13]. However, the 70–100 kyr lag in the recovery of climate and isotopic composition suggests that the PETM warming shifted terrestrial ecosystems into a long-lasting state of reduced carbon storage, and that the time needed for vegetation systems to recover from the perturbation and to re-establish their carbon sequestration potential may have strongly influenced the duration of the hyperthermal.

Vegetation functioning in response to global warming depends on the ability of plants to avoid stress conditions through migration or to

[1]Department of Earth and Planetary Sciences, ETH Zurich, Zurich, Switzerland. [2]Department of Environmental Systems Science, ETH Zurich, Zurich, Switzerland. [3]School of Geography, Earth and Atmospheric Sciences, University of Melbourne, Melbourne, Australia. [4]Department of Geology and Geophysics, University of Utah, Salt Lake City, Utah, USA. [5]NSF National Center for Atmospheric Research, Boulder, Colorado, USA. [6]Swiss Federal Institute for Forest, Snow and Landscape Research, Birmensdorf, Switzerland. ✉e-mail: jul.rogger@gmail.com

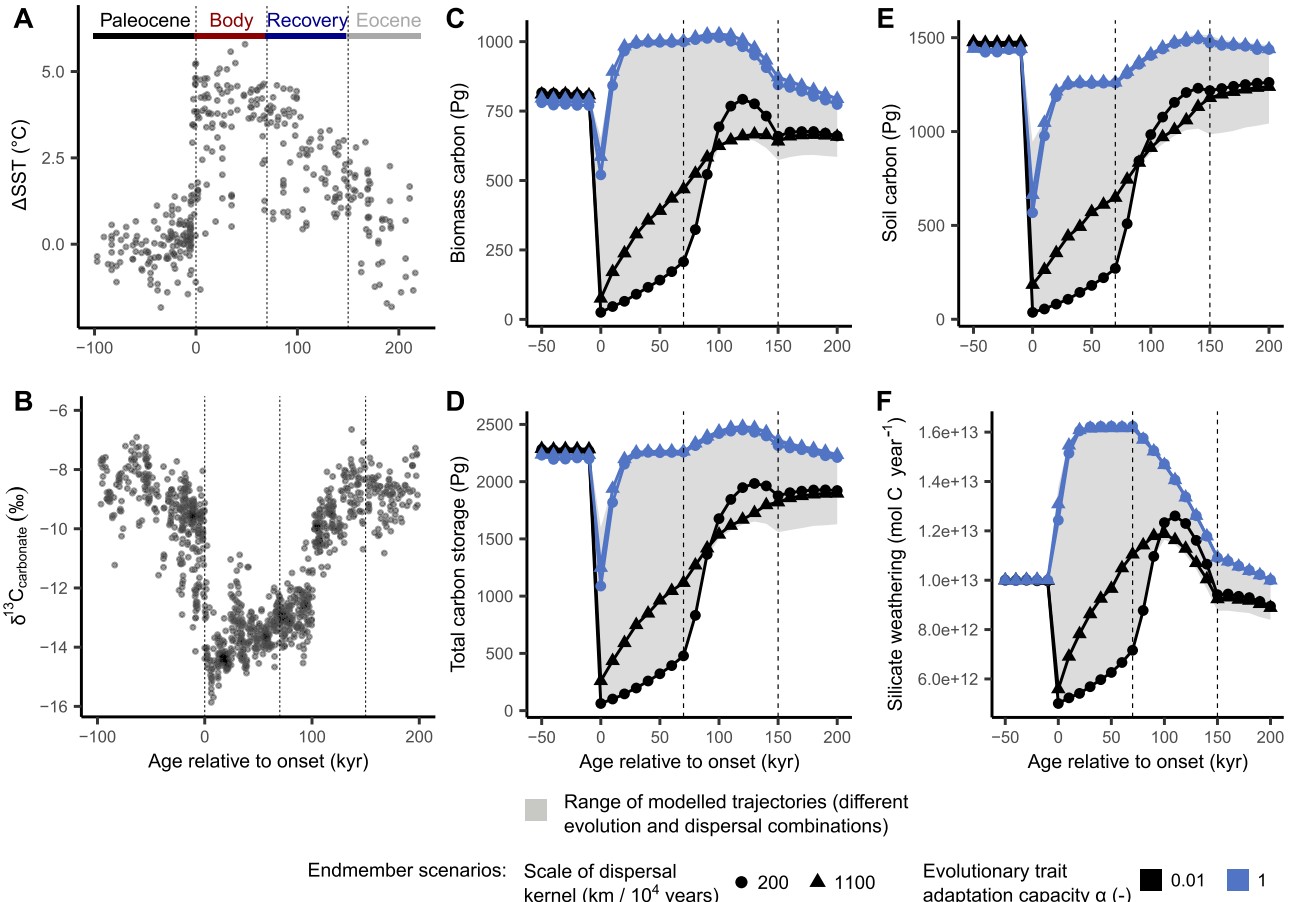

**Fig. 1 | Proxy-derived and modelled carbon cycle dynamics across the Paleocene−Eocene Thermal Maximum (PETM). A** Sea surface temperature changes (ΔSST) based on planktic foraminifera Mg/Ca and lipid biomarkers, plotted relative to pre-PETM average values, as compiled in ref. 97, based on data from refs. 98−100. **B** High-resolution $\delta^{13}C$ records of the PETM carbon isotope excursion measured on pedogenic carbonate nodules in the Bighorn Basin, Wyoming (USA). Data from ref. 101. **C** Modelled carbon storage in biomass (aboveground + belowground). **D** Total carbon storage in biomass and soil. **E** Soil carbon storage as a function of net primary productivity, carbon inputs and temperature-dependent heterotrophic carbon respiration. **F** Silicate weathering as a function of erosion, runoff, temperature and vegetation-mediated weathering enhancement. The dispersal value represents the scale parameter of a Weibull-based dispersal kernel with a shape parameter value of 1.75, from which dispersal distances are drawn randomly for each modelled plant and time step. The evolutionary adaptation capacity ($\alpha$) is given relative to the selection pressure, where a value of 1 indicates an immediate transition of traits towards the optimum trait combination predicted in a given environment (i.e., evolution of 100% of the optimum to current trait value difference). The grey-shaded area gives the range of trajectories observed in model simulations considering different combinations of vegetation dispersal and evolutionary trait adaptation between the indicated extreme endmember scenarios. The vertical dotted lines in (**C**, **D**, **E** and **F**) indicate the PETM recovery period. Figure produced using R[92] and the ggplot2 package[93].

prevail in the new environment through adaptation and evolution[14,15]. For selected fossil species, large dispersal distances have been observed in response to the PETM warming[5,16]. However, it is unclear whether a widespread migratory response and the resulting reorganisation of vegetation structures[6] were sufficient to attenuate the effects of the warming and to compensate for a potential loss of vegetation adaptation to the new environmental conditions. The evolutionary capacity of plants to adjust to new climates is expected to be limited in speed and magnitude, with climatic niches occupied by species remaining conserved on geologic timescales[17–21]. Slow evolution and a warming-induced adaptation lag could have reduced organic carbon sequestration during the PETM, and may have impacted additional vegetation-mediated carbon cycle feedback mechanisms, such as the biotic enhancement of $CO_2$-consuming terrestrial silicate weathering reactions[22–26].

Here, we assess how vegetation dispersal and evolution in response to the PETM affected the global carbon cycle and vegetation-mediated climate regulation. For this purpose, we developed a trait- and optimality-based biogeodynamical vegetation model, designed for continuous simulations of vegetation structure and functioning

over geologic timescales. In addition to global simulations, for three sites in the mid- to high latitudes, we compare simulated vegetation trends with functional trait reconstructions using nearest living relatives of fossil species observed in palynofloral records, as well as available macrofossil observations. Using model simulations and vegetation reconstructions, we investigate the possibility of a disrupted terrestrial carbon cycle due to a warming-induced vegetation adaptation lag.

The developed vegetation model combines plant physiology and allometry from common dynamic vegetation models[27,28] with eco-evolutionary optimality theory to predict plant trait evolution[29,30] and a model to consider vegetation dispersal[31]. A continuous and evolving trait space, combined with an explicit representation of dispersal dynamics, are the main differences to common plant-functional-type-based vegetation models[32]. Key traits included in the model are plant leaf longevity, growth form (height, carbon partitioning between tissues), phenology (deciduous or evergreen), and climatic niche. Plant functional traits are subject to carbon economic trade-offs. These include the leaf economics spectrum, describing a trade-off between leaf longevity and specific leaf area, and carbon costs of height growth,

which is driven by light competition[29,30,33,34]. The climatic niche trait limits a modelled plant's productivity potential under climatic conditions outside the usual habitat. Derived from the climatic variation observed within present-day biomes (Supplementary Fig. 1), the baseline model allows a maximum 10 °C deviation in the mean annual temperature, the maximum, or the minimum monthly average temperature from the usual habitat before a loss of productivity occurs. Mimicking natural selection, plant traits and climatic tolerances are assumed to evolve towards optimum adaptation to the local climate conditions, taken as the trait combination that maximises vegetation height and the net carbon gain, which is the carbon from photosynthesis that remains after accounting for carbon investments related to respiratory processes and tissue turnover. While evolutionary dynamics result in vegetation systems to naturally consist of many co-existing growth forms and trait combinations, the optimality prediction serves as a first-order approximation of the vegetation type that is likely dominant and stable in a given climatic environment[29,30]. Considering present-day climatic conditions, the model reproduces major vegetation patterns, including net primary productivity, biomass carbon storage, vegetation height and evapotranspiration (Supplementary Figs. 2–6). From the vegetation model output, we further derive the potential soil carbon storage, applying a steady-state model balancing net primary productivity carbon inputs and temperature-dependent heterotrophic soil carbon respiration[35]. Further, we estimate how the modelled vegetation dynamics could affect the efficiency of carbon drawdown through silicate weathering. Thereby, weathering rates are approximated as a function of erosion, runoff, temperature and a local vegetation enhancement factor depending on primary productivity. In areas of highest plant productivity and biomass, we apply a maximum six-fold weathering enhancement by vegetation, in the range of field- and laboratory-derived enhancement factors[22–26,36].

We evaluate the dynamics of terrestrial carbon cycling for different vegetation adaptation scenarios across the PETM. The capacity of vegetation to transition from a prevailing trait distribution to the predicted optimum traits under new climatic conditions depends on the rate of evolutionary trait adaptation and dispersal dynamics. Evolutionary trait adaptation is considered relative to the selection pressure, i.e., the difference between the optimum and the current trait values, where an adaptation rate parameter ($\alpha$) of 1 indicates near immediate evolution of traits towards the optimum. The rate of trait evolution of vegetation systems to maintain functioning under climatic changes is a major unknown both for past[37] and present[14,17,18] climatic shifts. In our simulations, we consider a large range from minimal evolution ($\alpha = 0.01$) to near immediate adaptation ($\alpha = 1$) to compare associated carbon fluxes and vegetation structures to geochemical and paleobotanical observations. To evaluate the effect of a migratory response of vegetation, dispersal distances of the modelled plants are drawn from a Weibull-shaped dispersal kernel with a varying scale parameter between simulations. Distances under the highest dispersal scenario (kernel scale parameter = 1100 km per 10 kyr) exceed estimated range shifts from PETM fossils (650 to 1500 km per 10 kyr)[5].

Pre-PETM and PETM climate boundary conditions for the vegetation modelling were derived from climate simulations using the Community Earth System Model, run at 680 and 1590 ppm atmospheric $CO_2$[6]. The applied climate forcing consists of 50 kyr pre-PETM climate conditions at an atmospheric $CO_2$ of 680 ppm, followed by a step change to 1590 ppm across one model time step of 10 kyr (PETM onset; average land surface warming of ≈ 8 °C), followed by a 70 kyr period at 1590 ppm $CO_2$ (the PETM body), then a gradual recovery back to 780 ppm $CO_2$ until 150 kyr after the PETM onset, and a final atmospheric $CO_2$ concentration of 680 ppm at the end of the simulation, 200 kyr after the PETM onset (Supplementary Fig. 7). The model evaluates vegetation dynamics at 10 kyr intervals

across the PETM, considering multiyear monthly average climate inputs at each time step. The monthly climate fields during the PETM recovery period are derived by linear interpolation from the available pre-PETM and PETM body climate simulations. Besides evaluating carbon cycling under different rates of evolutionary vegetation trait adaptation and dispersal in response to this reference warming scenario, we further test a range of additional warming scenarios with the same temporal warming trajectory, but maximum atmospheric $CO_2$ concentrations during the PETM body ranging between 800 and 1590 ppm, with monthly climate fields required by the vegetation model again being derived by interpolation from the available pre-PETM and PETM body climate model runs. Finally, the vegetation model was run with and without considering $CO_2$ fertilisation effects on plant productivity.

## Results

### Vegetation adaptation dynamics pace terrestrial carbon cycling

The modelled response of vegetation to the PETM warming affects the global dynamics of terrestrial carbon storage and silicate weathering (Fig. 1). Assuming a limited evolutionary capacity of vegetation to adapt functional and climatic traits to the environmental changes results in a reduction of terrestrial carbon storage following the PETM onset, potentially lasting as long as the body of the carbon cycle perturbation (70–100 kyr). A loss and lagged regrowth of the biospheric organic carbon storage is in agreement with geochemical records for the period, suggesting a release and reduced storage of biospheric carbon during the onset and body of the PETM, followed by an enhanced sequestration of isotopically light carbon during the termination of the carbon cycle and climatic perturbation[9,13]. A lagged organic carbon regrowth in the range of 1000 to 2000 Pg is modelled under scenarios of limited evolution. In contrast, when assuming fast trait and climatic niche adaptation or no limitation in climatic tolerances (Supplementary Fig. 8), a less severe initial drop in carbon sequestration is soon compensated by a rapid regrowth and overshoot in global biomass carbon compared with pre-PETM levels. The overshoot in biomass indicates that under a reduced or absent adaptation lag, the temperature, precipitation and atmospheric $CO_2$ conditions during the PETM body would support globally increased rates of primary productivity. Such a vegetation–climate equilibrium, however, contrasts with fossil plant and soil records, which point towards a complete turnover of vegetation systems and diminished carbon storage following the PETM warming[4–6,38].

The long duration of the recovery of carbon stocks is also observed in simulations considering the highest dispersal capacities inferred from the fossil record. Therefore, a migratory response may not have been sufficient to compensate for a loss of functional and climatic trait adaptation following the PETM environmental changes. Under a high dispersal scenario, carbon stocks are modelled to steadily recover from the perturbation, due to a continuous spatial redistribution of vegetation and a reduction of the adaptive lag. For a low dispersal scenario, there is a limited recovery of carbon stocks during the PETM body, followed by an acceleration during the PETM recovery period. The accelerated recovery under low dispersal occurs because modelled plants tend to remain close to the original habitat and rapidly exit the PETM stress state with the return of climatic conditions similar to those of the late Paleocene. Such carbon dynamics would be expected if plants managed to persist in local refugia throughout the climatic perturbation.

A vegetation adaptation lag additionally impacts carbon sequestration through a reduced vegetation-mediated enhancement of terrestrial silicate weathering. Under a scenario of slow vegetation recovery, the effects of a warmer and wetter climate on silicate weathering are temporarily offset by a reduced vegetation-mediated weathering enhancement following the PETM climatic perturbation. Due to the vegetation disruption, an overshoot in silicate weathering,

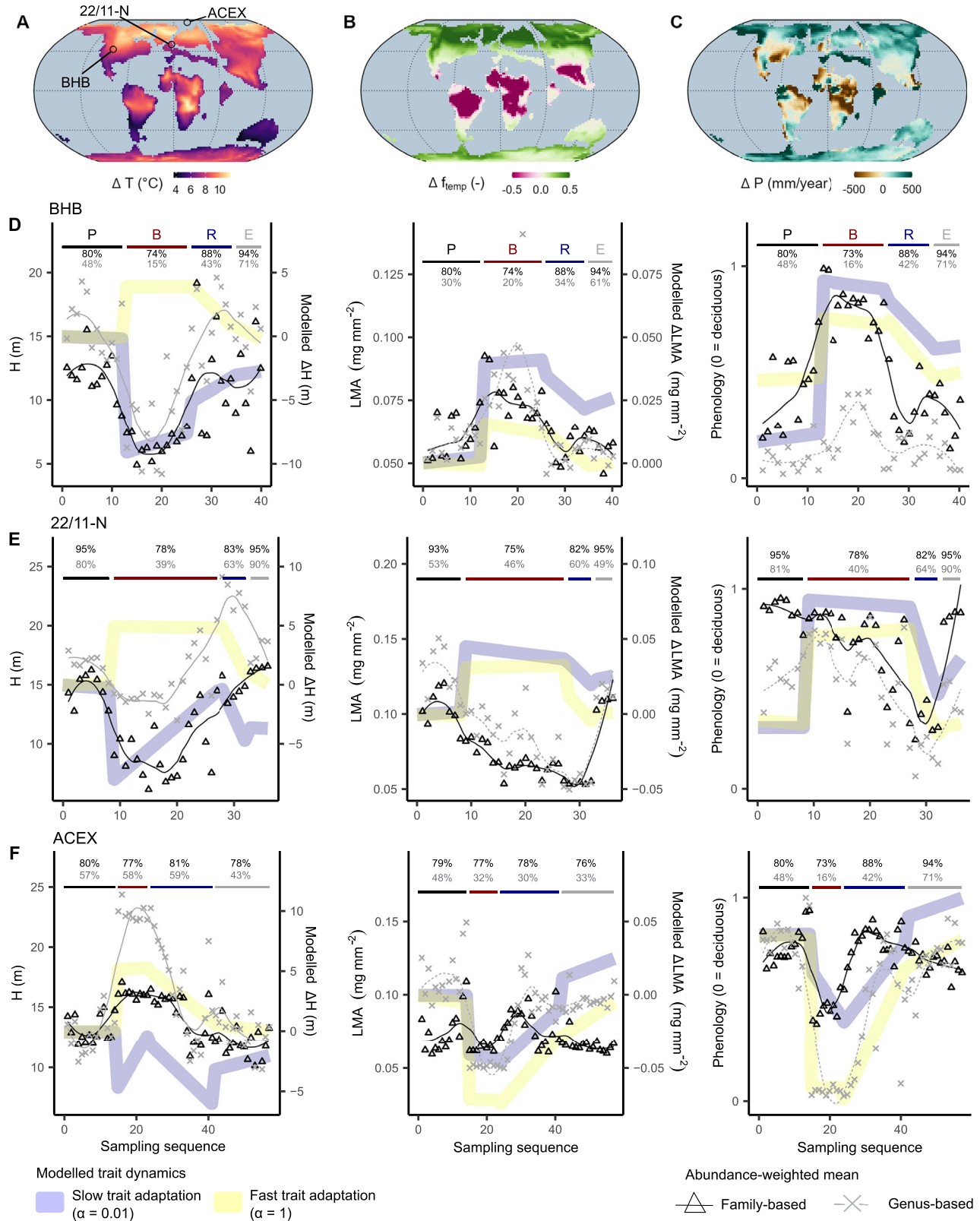

**Modelled trait dynamics**

- **Slow trait adaptation** (α = 0.01)
- **Fast trait adaptation** (α = 1)

**Abundance-weighted mean**

- △ Family-based
- ✕ Genus-based

and thus effective long-term climate regulation, may only occur with a lag of several tens of thousands of years.

## Palynofloral vegetation reconstructions

We use palynofloral records of species composition and abundance spanning the PETM to reconstruct vegetation trait changes at the two mid-latitude sites Bighorn Basin[6] and 22/11-N[7], as well as the high

latitude site ACEX (Arctic Coring Expedition)[8] (Fig. 2A). Thereby, we assign the recorded fossil species an average trait value of plant height, leaf mass per area (LMA) and phenology (deciduous or evergreen), as observed in modern relatives of the same genus or family. Due to continuous evolution and adaptation, absolute trait values may have changed between the fossil species and their living relatives today. However, by assuming that the relative ranking in trait values remained

**Fig. 2 | Reconstructed vegetation trait dynamics across the Paleocene–Eocene Thermal Maximum (PETM). A** Temperature (T) change between PETM body and pre-PETM climatic conditions and palynofloral sampling site locations Bighorn Basin (BHB), and sediment cores 22/11-N, and ACEX (Arctic Coring Expedition). **B** Change in the temperature limitation of photosynthesis ($f_{temp}$) between the PETM body and pre-PETM conditions, with positive values indicating a warming towards more optimal temperature for photosynthetic carbon assimilation of around 25 to 30 °C. **C** Changes in annual precipitation (P) between PETM body and pre-PETM climatic conditions. **D–F.** Palynofloral-based and modelled changes of vegetation height (H), leaf mass per area (LMA) and phenology (deciduous = 0 or evergreen = 1) across the PETM for (**D**). the BHB, (**E**). 22/11-N and (**F**). ACEX. Reconstructions are derived from trait values of the nearest living relatives of the fossil species observed at the three sites. Samples are in sequence and assigned to Paleocene (P), PETM body (B), PETM-recovery (R) and Eocene (E) time bins, due to the lack of a more detailed age control. Trends from the model represent the first and last value modelled for the respective time bins in the regions of the sampling sites under different evolutionary trait adaptation capacities (α). Model results are given as relative change over time, to account for the lack of a $CO_2$ fertilisation effect in the data (trait values derived from present-day trait values) and for the lack of an understory vegetation in the model. Model results under the highest dispersal scenario are shown. Percentages give the average fraction of the fossil species in the samples for which trait values exist for the nearest living relative at the family (black) or the genus (grey) level. The full range of trait values and abundances for all sites are presented in Supplementary Figs. 11, 12 and 13. More details on the sampling sites and the considered model regions are illustrated in Supplementary Fig. 14. Figure produced using R[92] and the ggplot2 package[93].

conserved across clades (Supplementary Fig. 9), we aim to approximate major trends in vegetation structural changes across the PETM.

For both mid-latitude sites Bighorn Basin and 22/11-N, the palynofloral reconstruction indicate a shift towards smaller plants (Fig. 2), likely associated with a reduced biospheric carbon storage (Supplementary Fig. 10). For the Bighorn Basin, the reconstructed reduction in vegetation height coincides with a reduced abundance of deciduous trees and an increase in average LMA (smaller and thicker leaves). The trait shifts are caused by a dominance of tropical to subtropical evergreen palms (i.e., *Serenoa*, *Pseudophoenix*) and ferns (i.e., *Pityrogramma*, *Acrostichum*, *Cibotium*, *Salvinia*, *Anaemia*) during the PETM body, while tall-growing trees of a reduced average LMA and likely more frequent deciduousness (i.e., *Metasequoia*, *Platanaceae*, *Betulaceae*, *Juglandaceae*) only reappear during the PETM recovery period. The vegetation shifts in the Bighorn Basin suggest a dominance of more conservative, but likely more robust plant strategies concerning potential heat and drought conditions during the PETM body[39]. Similarly, at the site 22/11-N, evergreen palms and ferns (i.e., *Schizaeaceae*, *Polypodiaceae*) and small wetland floras (i.e., *Typhaceae*) dominate the playnofloras of the PETM onset and body, resulting in a reduction of the average vegetation height. Mixed conifer-broadleafed vegetation (i.e., *Pinaceae*, *Metasequoia*, *Betulaceae*, *Juglandaceae*) reappears during the PETM recovery period. Compared to the Bighorn Basin, a less drastic height shift is reconstructed due to the persistence of some conifer and broadleaved trees throughout the perturbation.

The palynofloral reconstructions at both mid-latitude sites are best reproduced in the model simulations that consider a limited evolutionary capacity of vegetation to adapt functional and climatic traits following the PETM onset, resulting in a primarily dispersal-driven vegetation response. The loss of local adaptation results in reduced productivity, carbon sequestration and height growth during the body of the PETM. The combination of simulations and palynofloral records therefore support the possibility of a warming-induced adaptation lag, resulting in the dominance of robust, but more conservative plant strategies following the PETM onset, until more acquisitive strategies (i.e., tall, deciduous trees of low LMA) slowly return with time. Importantly, in contrast to the palynofloral reconstructions, simulations that do not consider an adaptation lag indicate a theoretically possible increase in vegetation height and carbon sequestration potential at both the Bighorn Basin and 22/11-N sites after the onset of the PETM warming. A loss of vegetation adaptation may therefore have offset environmental conditions in favour of increased primary productivity, causing more robust but less productive plant strategies to dominate the local vegetation. An additional indication of a limited evolutionary trait and climate adaptation capacity is the high degree of species turnover observed at the sites, with only a few species persisting throughout the climate excursion by either enduring the new climate or adjusting to it.

A mismatch is observed between modelled and reconstructed LMA for the 22/11-N site following the PETM onset. The model predicts

an initial shift towards increased LMA that is not observed in the data, whereas both reconstructions and the slow adaptation model predict a subsequent decreasing trend during the PETM body. The mismatch in the initial shift is likely caused by the model not explicitly representing the herbaceous understory vegetation that in terms of abundance, dominates the PETM-body palynoflora of 22/11-N. Instead, the optimality-driven model only captures the plant growth form that is most competitive regarding carbon sequestration and height growth. The model predicts the presence of 10–15 m evergreen trees of high LMA during the PETM body. This is in the range of growth forms reconstructed for the 22/11-N site from the palynofloral records, although at a reduced abundance (Supplementary Fig. 12). In both cases, the reconstructed and modelled reduction of height growth during the PETM warming suggests a reduced carbon sequestration at the site, supporting an adaptive lag and disrupted vegetation state.

We observe contrasting patterns for the high-latitude site ACEX, where simulations and palynofloral reconstructions suggest an increase or fast recovery of vegetation height and biomass carbon storage following the onset and during the body of the PETM. Subsequently, a decrease in the vegetation height is observed during the termination of the hyperthermal and the associated climate cooling during the PETM recovery period. The trait trends are driven by a replacement of pre-PETM mixed conifer-broadleaved forests with abundant needle-leaved taxa (i.e., *Pinus*, *Picea*) by broad-leaved swamp forests (i.e., *Juglandaceae*), including some rare subtropical evergreen palms (i.e., *Arecaceae*) and deciduous tropical trees (i.e., *Bombacoideae*) during the body of the climate perturbation. These reconstructed and modelled trends indicate that increased temperatures (Fig. 2B), longer growing seasons, sufficient precipitation, and increased atmospheric $CO_2$ levels may have enabled a fast recovery or prevented negative warming effects for vegetation systems in the high latitudes.

## Comparison with macrofossil observations
The palynofloral trait reconstructions rely on trait conservatism through time and only capture trait changes driven by turnover in species composition. A direct comparison of model simulations with paleo-functional trait observations, and an assessment of potential evolution-driven changes in the trait ranking among the occurring species (Supplementary Fig. 9), is limited by the availability of preserved macrofossils. The Bighorn Basin is the only locality with a stratigraphically dense record of plant macrofossils throughout the PETM[40]. Macrofossils from the Bighorn Basin generally support the vegetation dynamics derived from the palynofloral record, indicating a replacement of mixed conifer-deciduous temperate forest taxa (e.g., *Platanaceae*, *Betulaceae*, *Cupressaceae*) with tropical and subtropical taxa (e.g., *Fabaceae*, *Arecaceae*, *Hernandiaceae*), likely resulting in a more open vegetation and an increased abundance of heat- and aridity-adapted species[4,39,40]. No direct estimates of height and biomass are available, but organic carbon measurements from terrestrial sediments and fossil soils indicate a major reduction in organic carbon

contents during the PETM[41,42] (Supplementary Fig. 15). Depleted soils and reduced carbon inputs from plant productivity are consistent with the reduced productivity and height growth derived in the palynofloral reconstruction and the slow evolution model (Fig. 2). Petiole width-based estimates of LMA for dicot angiosperms in the Bighorn Basin between the late Paleocene and early Eocene show no significant within-species LMA variations[43,44], which supports limited evolution-driven trait changes on PETM timescales. Further, similar as in the palynofloral reconstruction, the PETM indicates a marked increase in the abundance of high LMA species (i.e., *Fabaceae*) (Supplementary Fig. 15).

For the high latitudes, the temporal dynamics reconstructed and modelled for the site ACEX during the PETM resemble the latitudinal vegetation gradient observed in macrofossils from the late Paleocene and early Eocene Canadian Arctic. Despite the extreme photo-periodicity in these latitudes, macrofossils indicate forests composed of conifer and broadleaved (low LMA) deciduous trees, as well as some tropical palms in the lower Arctic latitudes[45,46]. Leaf physiognomic analyses indicate growth conditions similar to today's temperate forests or temperate rainforests[45]. Fossil tree stumps from the late Paleocene show diameters that indicate tree heights well beyond 20 m, in line with the height growth potential derived from the palynofloral reconstruction[47]. While there is no macrofossil assemblage available that confidently captures the onset, body and recovery phases of the PETM, there is currently no Canadian Arctic late Paleocene to early Eocene macrofossil site indicating a pronounced loss of productivity or diversity during the period[45,48]. This is supported by limited changes observed in the organic geochemical composition of coals deposited in the Canadian Arctic across the PETM[49], and agrees with the stable vegetation reconstruction derived from the palynofloral species composition at ACEX.

### Warming scenarios and $CO_2$ fertilisation

Modelled vegetation functioning during the PETM depends on the magnitude of experienced surface warming. We tested the vegetation response to different land surface warming scenarios, from an average of 1 °C to 8 °C (PETM body $CO_2$ concentrations between 800 and 1590 ppm), with the warmer endmember corresponding to estimates for the PETM[2] (Fig. 3A, B). From our simulations, we estimate a threshold around 4 °C, beyond which a widespread migratory response vegetation is not able to compensate negative warming impacts, resulting in an adaptation lag and a reduction of vegetation productivity levels. Warming beyond this threshold results in non-analogous climatic environments, making vegetation-mediated carbon sequestration primarily dependent on the capacity of plants to adapt traits through evolution and to occupy climatic niches well outside their usual habitat. Below this warming threshold, no adaptation lag is observed, and increased atmospheric $CO_2$ concentrations and warmer climates promote increased levels of primary productivity.

We further find that the temporal dynamics of terrestrial carbon storage depend on the vegetation's ability to use increased atmospheric $CO_2$ levels for photosynthesis during the PETM climatic perturbation (Fig. 3C–E). The $CO_2$ fertilisation effect reduces the water loss per unit of carbon taken up for photosynthesis, increasing the vegetation's resilience to drought and heat. Our simulations indicate that any limitation in the $CO_2$ fertilisation effect strongly reduces the vegetation's carbon assimilation potential, and prolongs the lag in the recovery of terrestrial carbon stocks and productivity. A limitation in the $CO_2$ fertilisation effect could have been an additional consequence of a loss of local adaptation following the PETM climatic perturbation, resulting in the inability of plants to efficiently acquire and utilise local water and nutrient resources for photosynthesis. The importance of $CO_2$ fertilisation to maintain productivity and biomass is particularly pronounced in the mid-latitudes.

## Discussion

A regional loss of vegetation functions following the 5–6 °C PETM global warming may have strongly shaped the climate and carbon cycle dynamics of the hyperthermal event. Particularly in the mid-latitudes, our simulations and vegetation reconstructions based on nearest living relatives of PETM palynofloras indicate a reduced terrestrial carbon sequestration and a dominance of robust but less productive vegetation systems following the perturbation. Considering the >100 kyr timescale for complete recovery from the event, our results indicate limits in the ability and speed of vegetation to respond to abrupt climate warming, with negative implications for the functioning of terrestrial carbon sinks in response to present-day climate change.

We tested the effect of the two predominant vegetation responses to environmental change: migration and adaptation[14,15]. A migratory response of plants is the dominant feature observed in paleobotanical records of the PETM[5,6]. Range shifts of 650–1500 km within approximately 10 kyr have been reconstructed[5], which is comparable to postglacial dispersal estimates[50,51]. Despite the consideration of such dispersal rates, our simulations indicate that the PETM may have exceeded a threshold beyond which a redistribution of vegetation was insufficient to compensate for a loss of vegetation adaptation to the changing environmental conditions. Consequently, vegetation functioning may have been limited by the capacity of plants to adapt functional traits and climatic tolerances through evolution. Compared with migration, adaptive evolution is generally considered a much slower response mechanism. Phylogenetic estimates of climatic niche changes in plants through evolution are often reported to be within the range of a few degrees per million year[17,18,20], considerably slower than the PETM warming rates (5–6 °C over <20 kyr). The evolutionary adaptation potential can be constrained by limited genetic variation and heritability regarding the relevant functional and climatic traits[52], evolutionary trade-offs between traits under selection[53], or genetic bottlenecks during migration[15,54]. The adaptation process may further be delayed by changing abiotic and biotic disturbances to vegetation systems, such as increased insect herbivory[43].

Our simulations considering a limited ability of vegetation to adapt to the PETM climatic changes reproduce the pronounced changes in vegetation traits reconstructed based on the species turnover in PETM palynofloras and limited macrofossil observations. In mid-latitudes (Bighorn Basin, 22/11-N), the simulated and reconstructed dynamics suggest a limited vegetation fitness despite evidence for fairly long-distance migration, resulting in more conservative and less productive trait combinations due to an eco-evolutionary adaptation lag. In simulations that assume near-immediate evolutionary adaptation of vegetation to climatic changes, the productivity during the body of the climatic excursion increased at the considered mid-latitude sites and globally compared to pre-PETM levels. Such an increase in the global productivity potential has also been observed in steady-state vegetation model simulations for the period that did not consider eco-evolutionary adaptation constraints[55]. Importantly, an eco-evolutionary adaptation lag in the presented simulations does not imply a loss or extinction of plant species, which have been limited during the PETM[4]. An adaptation lag rather indicates that the dispersal-driven redistribution of plants results in a temporary productivity loss, and a vegetation state that in terms of carbon sequestration remains below the theoretical potential. The difference between potential and realised carbon sequestration emphasises the need to understand biological adaptation dynamics as a driving factor in the Earth system response to warming. Vegetation not only responds to environmental changes but actively shapes global carbon and climate dynamics by determining the efficiency of carbon cycling. We observe a particular sensitivity of mid-latitude vegetation functioning to warming, in contrast to stable or increasing productivity levels in the high latitudes. Therefore, a

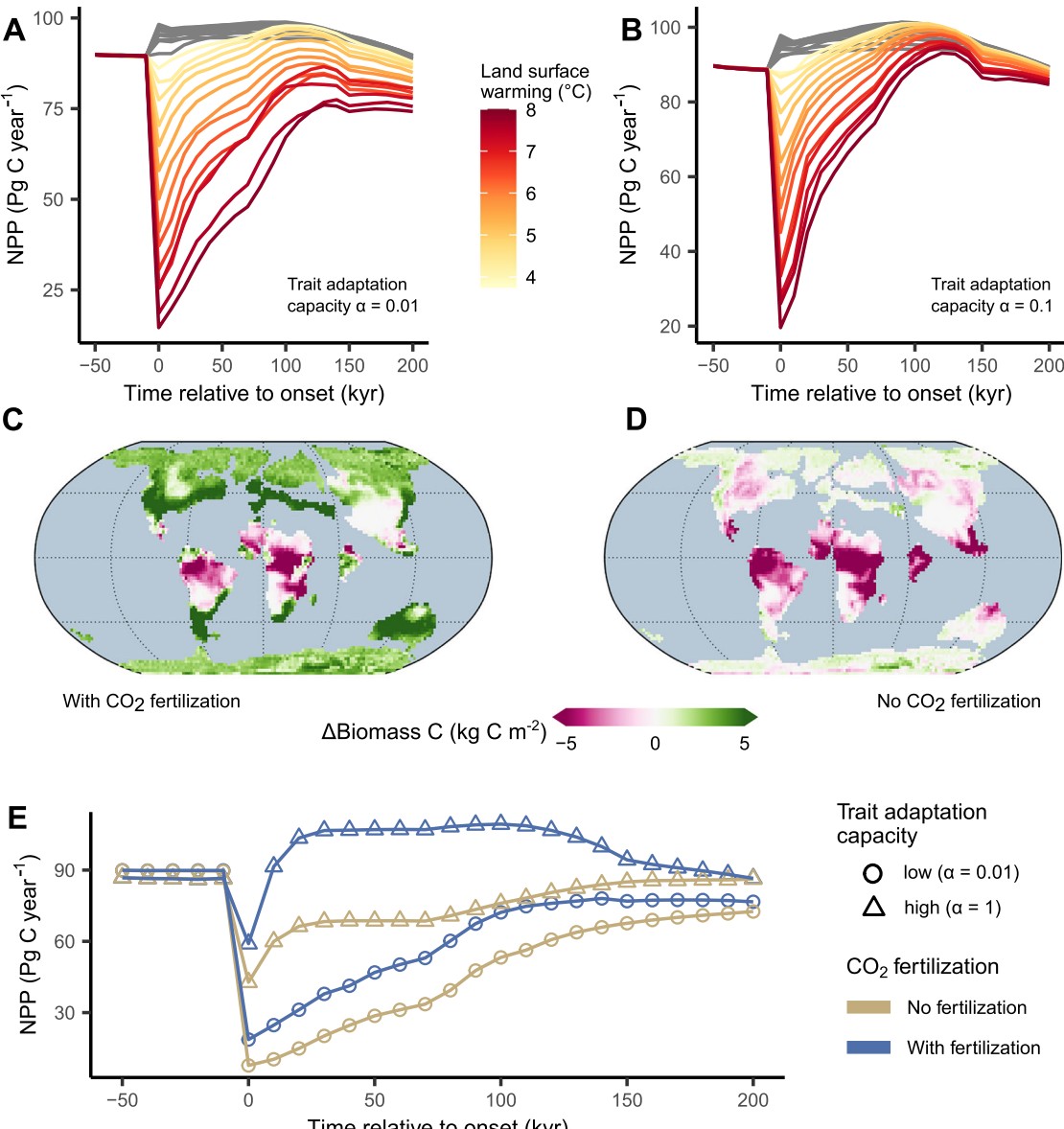

**Fig. 3 | Sensitivity to different warming scenarios and CO₂ fertilisation effects.** Trajectories of net primary productivity (NPP) for different land surface warming scenarios: **A** considering a functional and climatic trait adaptation capacity ($\alpha$) of 0.01 (= 1% per timestep of 10 kyr), and (**B**) considering a trait adaptation capacity of 0.1. Simulations under the highest dispersal scenario are shown (scale of dispersal kernel = 1100 km per $10^4$ years). Grey trajectories indicate land surface warming scenarios below 4 °C, showing no negative impact on global NPP levels. Changes in vegetation biomass (aboveground + belowground) between Paleocene-Eocene Thermal Maximum (PETM) body and pre-PETM conditions (**C**) with and (**D**) without CO₂ fertilisation for a vegetation model implementation considering a high dispersal and unlimited evolutionary adaptation capacity. **E** Temporal dynamics of NPP across the PETM with and without considering a potential CO₂ fertilisation effect and a low or high trait adaptation capacity of vegetation. Figure produced using R[92] and the ggplot2 package[93].

better temporal and spatial coverage of palynofloral- and macrofossil-based vegetation reconstructions in the mid-latitudes is of particular importance to understand and constrain the vegetation-mediated carbon sequestration trajectory during the PETM.

A loss and slow recovery of vegetation carbon stocks and silicate weathering rates in simulations considering a limited evolutionary climate adaptation capacity are in agreement with PETM carbon flux estimates based on geochemical proxies. Proxy analyses indicate that around 5800 Pg of carbon were necessary to induce the CO₂ changes observed across the PETM and that the release of isotopically light organic carbon, likely driven by a carbon cycle feedback response to an initial warming, may have contributed substantially to the total carbon release[1]. Further, based on the observed rates in the recovery of $\delta^{13}$C signatures during the PETM recovery, it was estimated that a

sequestration of 2000–2500 Pg of organic carbon contributed to the termination of the event[9,12,13]. With an estimated total carbon pool of 2000–2500 Pg in vegetation and soils, our simulations indicate that a loss and lagged regrowth of 50% or more of this carbon pool may have contributed substantially to the global carbon cycle dynamics. We show that the vegetation disruption could additionally have reduced the efficiency of chemical weathering of silicate minerals, causing a delay of up to 50 kyr in the overshoot of carbon consumption associated with silicate weathering to rates above pre-PETM levels. A lag in the delivery of inorganic carbon from continental weathering is in line with the delay in the carbonate accumulation overshoot and a deepening of the carbonate compensation depth observed in marine sediments compared with biogeochemical model predictions for the period[56,57]. While a biospheric carbon loss and a reduced terrestrial

carbon sequestration may have caused a prolonged warming during the PETM, the combined recovery of vegetation systems and weathering rates towards the end of the PETM body could have triggered a range of carbon cycle feedbacks contributing to the termination of the hyperthermal. Increased delivery of weathering-derived nutrients to the ocean promotes marine primary productivity and organic carbon burial[58–60]. With the recovery of vegetation and soil carbon stocks, elevated rates of physical erosion during the PETM climatic conditions[58,61] could have promoted the transport and burial of terrigenous organic matter in the ocean. Our results highlight the importance of studying the eco-evolutionary response of vegetation to warming as a tightly coupled system with climate and the geologic carbon cycle. With the prescribed climate boundary conditions in the presented simulations, we did not take into account a dynamic feedback between vegetation and the atmospheric carbon balance through the loss and delayed sequestration of carbon, nor direct biophysical interactions with the climate system, such as changes in surface albedo and water cycling. Taking into account such dynamic interactions will be essential to establish a better understanding how a loss of vegetation-mediated climate regulation after an abrupt warming, as suggested by our simulations and palynofloral analyses, may trigger positive feedback dynamics in the climate and carbon system, determining the magnitude and duration of climate perturbations.

Our PETM simulations and vegetation reconstructions have implications regarding the capacity of terrestrial carbon sinks to mitigate anthropogenic climate change. For the PETM, we observe a critical threshold in land surface warming of around 4 °C beyond which the positive effects of a warmer climate and increased atmospheric $CO_2$ concentrations are offset by a loss of adaptation of vegetation systems. With warming of this magnitude, vegetation needs to adapt to environments well outside the original habitat to persist, potentially resulting in reduced carbon sequestration. For anthropogenic climate change, this absolute warming threshold may lay higher due to the cooler background climate state. Thus, the positive effects of warming and $CO_2$ fertilisation observed in the high latitudes during the PETM potentially extend to lower latitudes, and fewer regions may suffer from temperatures reaching critical thresholds for photosynthesis (Fig. 2B). However, the warming today occurs at a much higher speed, with current rates of carbon release exceeding those during the PETM onset likely by around an order of magnitude[62,63]. Compared to the warming imposed in the presented modelling (1–8 °C over a 10 kyr time step), current warming rates imply substantially less time for species to adapt and disperse. While rates of evolutionary adaptation may be slow both on the timescales of the PETM and current climate change, today, dispersal limitations may become particularly important in causing an adaptive lag. Estimates of vegetation migration rates needed to track climate change (>1000 m year⁻¹) are considerably higher than observed rates in fossil records for the PETM or postglacial periods[50,64–66]. Further, dispersal capacities are expected to decrease in the future due to increasing land fragmentation[67], as well as the loss of seed dispersal vector species[68]. Current vegetation models further assume a strong positive $CO_2$ fertilisation effect on vegetation productivity. There are, however, indications that the extent of this fertilisation effect may decline due to emerging nutrient limitations or climate-induced stresses, including more frequent heat and drought periods[69–72]. Our simulations show that a slow speed of the vegetation response to an abrupt warming of equivalent magnitude to the PETM, through limited dispersal or a loss of local adaptation, may strongly reduce vegetation resilience to stress and impact long-term vegetation functioning and carbon sequestration.

## Methods
### Vegetation model
Vegetation modelling was conducted using the newly developed TREED model (TRait Ecology and Evolution model over Deep time,

version 0.1). In addition to the summary of the model provided here, an extended description of all model equations and parameters is given in the Supplementary Methods (SM). TREED is a trait- and optimality-based vegetation model designed to conduct continuous vegetation simulations over geologic timescales. Based on concepts of eco-evolutionary optimality[29,30], the model considers variable, continuous and evolving functional and climatic trait combinations of modelled plants over time. Following the scheme in Supplementary Fig. 16, the model consists of four modules executed in succession.

**Initialisation, allometric and functional relationships.** The model is initialised with one reference plant per terrestrial grid cell, characterised by six key traits that describe the vegetation structure and functioning of the local vegetation. The key traits include: plant height ($H$), mass of the leaf carbon pool ($C_{leaf}$), phenology (deciduous or evergreen), fine root to leaf carbon ratio ($r{:}s$), leaf longevity ($a_{ll}$), and climatic niche. The climatic niche describes the climatic environment to which the modelled plant is best adapted, characterised by the temperature of the coldest month ($T_{min,veg}$), the temperature of the warmest month ($T_{max,veg}$), and the mean annual temperature ($T_{mean,veg}$) of this environment. Large differences between local climatic conditions and a modelled plant's climatic niche, for example, after climatic changes or migration, can result in a reduction in productivity, as described below. At model initialisation, the climatic niche of the modelled reference plant in a grid cell is derived from local climatic conditions, assuming an optimal adaptation. Using the mentioned key traits and allometric and functional relationships, several other plant characteristics relevant to vegetation structure and functioning are derived. One key trait relationship is the dependency between a modelled plant's specific leaf area (SLA) and leaf longevity ($a_{ll}$), according to (all model parameters and variables, including units, are listed in Supplementary Tables 1 and 2):

$$\text{SLA} = (2 \cdot 10^{-4}) \cdot \frac{1}{\text{DM}_c} \cdot 10^{2.25 - 0.5 \cdot \log(a_{ll} \cdot 12)} \tag{1}$$

This functional relationship represents a generalised form of the relationship used for needle-leaved and broadleaved plants in the LPJml4 dynamic vegetation model[28]. It represents a carbon-economic trade-off between a high SLA, promoting a large leaf area and thus a high carbon capture potential, and high annual leaf building costs due to a low leaf longevity[33]. $\text{DM}_c$ is an assumed average dry matter carbon content of 0.47 g C per g of dry matter, following[28]. SLA and carbon assimilation potential are related through the leaf area index (LAI), which ultimately determines the potential to capture light for photosynthesis:

$$\text{LAI} = \frac{C_{leaf} \cdot \text{SLA}}{\text{CA}} \tag{2}$$

where CA is the "crown area", representing the surface area occupied by a modelled plant and that scales with plant height (SM).

Several additional allometric relationships are used to describe plant geometry and the carbon partitioning between different plant carbon pools ($C_{sapwood}$, $C_{heartwood}$, $C_{fineroot}$, $C_{coarseroot}$, $C_{leaf}$) (SM). Given that there are no pre-defined plant functional types, the allometric relationships are generalised for all modelled plants. All parameters related to geometry and carbon partitioning are calibrated using present-day vegetation height and biomass data.

**Carbon balance.** Based on a reference plant's traits, the carbon balance module estimates photosynthetic carbon assimilation, maintenance and growth respiration, as well as average carbon investments for tissue turnover at a location. Photosynthesis is modelled based on the Farquhar photosynthesis model, considering generalisations for

global modelling applications[27,28,73–76]. Based on ref. [28], the absorbed photosynthetically active radiation (APAR) in each terrestrial grid cell depends on the incoming shortwave radiation and the light capture potential of the local reference plant, according to:

$$APAR = PAR \cdot \left(1 - e^{-k \cdot LAI}\right) \cdot \alpha_{leaf:stand} \cdot \Phi_{nichestress} \quad (3)$$

where half of the downwelling surface shortwave radiation (RSDS) is assumed to be photosynthetically active radiation (PAR = $0.5 \cdot$ RSDS) and $\alpha_{leaf:stand}$ is a leaf to stand level photosynthesis scaling parameter. $k$ is a light extinction coefficient, which is 0.6 for deciduous plants with $a_{ll} < 1$, 0.4 for plants with $a_{ll} > 4$, and 0.5 in all other cases. $\Phi_{nichestress}$ is an introduced stress factor that represents a potentially reduced carbon capture potential for plants in climatic environments to which they are not adapted. It is calculated as:

$$\Phi_{nichestress} = \min(\Phi_{heat}, \Phi_{cold}, \Phi_{mean}) \quad (4)$$

with

$$\Phi_{heat} = \exp(-k_{nichebreadth} \cdot (T_{max,local} - T_{max,veg})^2) \quad (5)$$

$$\Phi_{cold} = \exp(-k_{nichebreadth} \cdot (T_{min,local} - T_{min,veg})^2) \quad (6)$$

$$\Phi_{mean} = \exp(-k_{nichebreadth} \cdot (T_{mean,local} - T_{mean,veg})^2) \quad (7)$$

where $k_{nichebreath}$ describes the impact of a deviation of the local warmest month temperature ($T_{max,local}$), local coldest month temperature ($T_{min,local}$), and local mean annual temperature ($T_{mean,local}$) from the plant's climatic niche ($T_{max,veg}$, $T_{min,veg}$ and $T_{mean,veg}$, respectively). Three $k_{nichebreath}$ values are tested, i.e., 0, 0.03 and 0.10, corresponding to no, intermediate and strong limitation of primary productivity in new climatic environments, respectively (Supplementary Fig. 8). The default intermediate niche breadth is derived from the temperature variations observed within present-day biomes (i.e., regions with similar vegetation types) (Supplementary Fig. 1) and considers a maximum 10 °C deviation in local mean, warmest month or coldest month temperature from the plant's climate niche before a loss of productivity occurs. Based on APAR, photosynthesis is calculated as the minimum of light-limited and Rubisco-limited photosynthesis (SM). The model has the form of a "light-use efficiency" model, with PAR, temperature, day length, water availability and canopy conductance as the major modulating factors.

The rate of carbon assimilation is subject to a closed water balance, where the water loss from vegetation during photosynthesis (actual evapotranspiration, AET) cannot exceed the water supplied by precipitation. Following[27], AET is calculated as:

$$AET = \min(E_{supply}, E_{demand}) \quad (8)$$

where $E_{supply}$ is the monthly average precipitation rate, and a monthly average $E_{demand}$ is calculated following[77]:

$$E_{demand} = E_{eq} \cdot \frac{\alpha_m}{1 + \frac{g_m}{g_c}} \quad (9)$$

where $\alpha_m$ ($= 1.391$) is a Priestley-Taylor coefficient, $g_m$ ($= 3.26$ mm/s) is a scaling factor[28], $g_c$ is the canopy conductance, and $E_{eq}$ the equilibrium evapotranspiration rate (SM). Canopy conductance and photosynthesis are related by[27,28]:

$$g_c = \frac{1.6 \cdot A_{dt}}{p_a \cdot (1 - \lambda)} + g_{min} \quad (10)$$

where $A_{dt}$ is the daily net daytime photosynthesis (SM), $g_{min}$ is the minimum canopy conductance that occurs due to non-photosynthesis-related processes, and $p_a$ is the ambient partial pressure of $CO_2$. $\lambda$ describes the leaf–atmosphere water and carbon exchange and equals $\lambda_{max}$ (0.8) under non-water-limited conditions. Water limitation occurs if $E_{supply} < E_{demand,max}$, where $E_{demand,max}$ is calculated with Eq. (9), assuming a $\lambda$ value of $\lambda_{max} = 0.8$. In case of a water limitation, $\lambda$ is solved for to fulfil Eq. (10) and to obtain a closed water balance ($E_{demand} = E_{supply}$). The adjustment of $\lambda$ results in an adjustment of the photosynthetic rate and the canopy conductance accordingly.

Maintenance respiration rates for metabolically active carbon pools ($C_{leaf}$, $C_{sapwood}$, $C_{fineroot}$) depend on temperature and tissue-specific C:N ratios for sapwood and fineroot carbon, and on the maximum Rubisco capacity for leaves (SM). A fixed fraction of the modelled plant's net carbon assimilation is assumed to go into growth respiration[27,28], resulting in:

$$NPP = (1 - r_{gr}) \cdot (GPP - R_{leaf} - R_{sapwood} - R_{fineroot}) \quad (11)$$

where $r_{gr}$ is the fraction that goes into growth respiration and GPP (gross primary productivity) is the annual sum of the monthly average rates of daytime gross photosynthesis. $R_{leaf}$, $R_{sapwood}$ and $R_{fineroot}$ are the leaf, sapwood and fineroot maintenance respiration rates, respectively.

The model does not explicitly resolve the establishment, growth and mortality of individual plants. To evaluate whether a certain vegetation form is sustainable in a location, an annual average carbon balance is calculated, considering the average carbon costs of tissue turnover derived from the modelled reference plant's characteristics and carbon pools. The carbon balance ensures that the carbon capture potential of the reference plant is large enough to build and maintain the modelled plant characteristics. For sapwood, heartwood and coarse root carbon pools, the carbon costs of tissue turnover ($\tau_{sapwood}$, $\tau_{heartwood}$ and $\tau_{coarseroot}$, respectively) are calculated based on tissue-specific turnover times ($f$) and heat damage (HD) and frost damage (FD) indices, accounting for higher carbon costs of turnover in extreme environments (maximum doubling; SM), resulting in:

$$\tau_{sapwood} = C_{sapwood} \cdot f_{sapwood} \cdot (1 + HD + FD) \quad (12)$$

$$\tau_{heartwood} = C_{heartwood} \cdot f_{heartwood} \cdot (1 + HD + FD) \quad (13)$$

$$\tau_{coarseroot} = C_{coarseroot} \cdot f_{coarseroot} \cdot (1 + HD + FD) \quad (14)$$

The carbon costs of leaf turnover depend on leaf longevity:

$$\tau_{leaf} = \begin{cases} C_{leaf} \cdot \frac{1}{a_{ll}} & \text{evergreen} \\ C_{leaf} \cdot 1 & \text{deciduous} \end{cases} \quad (15)$$

Further, for leaf longevities longer than a year, the carbon costs of fineroot and leaf turnover are assumed to be proportional[28]:

$$\tau_{fineroot} = \begin{cases} C_{fineroot} \cdot \frac{1}{a_{ll}} & a_{ll} > 1 \\ C_{fineroot} \cdot 1 & a_{ll} \leq 1 \end{cases} \quad (16)$$

Subsequently, the annual carbon balance can be calculated as:

$$NCG = NPP - (\tau_{leaf} - \tau_{sapwood} - \tau_{heartwood} - \tau_{coarseroot} - \tau_{fineroot}) \cdot \frac{1}{CA} \quad (17)$$

where NCG is the net carbon gain, i.e., the carbon available from photosynthesis after accounting for the carbon costs of maintenance

and growth, respiration and tissue turnover. This carbon is available to the plant for reproduction, defence, nutrient acquisition and other processes relevant to survival. NCG is thus considered a fitness measure to compare the suitability of plant characteristics in a given environment.

**Trait optimisation.** Over time, environmental changes result in selection pressures that drive changes in vegetation structure and functioning. Following optimality principles[29,30], it is assumed that natural selection results in trait combinations evolving towards increasingly optimal adaptation to the local environment. To imitate this selection process, an optimisation algorithm is employed at every time step and for every terrestrial grid cell to predict the trait combination that maximises NCG (considering radiation, temperature and precipitation as environmental inputs). For this, a differential evolution algorithm for multi-parameter optimisation problems (DEoptim) is employed[78]. The optimisation predicts the optimum $C_{leaf}$, phenology and $a_{ll}$. In environments with more than 3 months with average temperatures below 3 °C (considered the necessary condition for a deciduous phenology), the optimisation procedure evaluates whether deciduousness results in a larger NCG compared with an evergreen phenology. In this model step, the maximum potential plant height for a given trait combination is also evaluated, defined as the height at which the ratio of NCG to NPP is approximately 0.2, following the assumption that this is the minimum NCG necessary to ensure reproductive success.

**Adaptation and dispersal.** At every time step of the model (10 kyr), the traits of the modelled plants evolve towards the predicted local optimum trait combination from the optimisation procedure. The rate of trait adaptation per time step is a user-defined adaptation rate ($\alpha$), which is varied between model simulations in the range from 0.01 to 1 to estimate the effect of a low to high ability of vegetation to adapt to new environments through trait evolution. Thereby, an adaptation rate of 1 indicates a trait change per time step that amounts to 100% of the difference between the predicted optimum and the current trait value, whereas an $\alpha$ of 0.01 represents an evolutionary trait change amounting to 1% of the difference. In the present model, the traits $C_{leaf}$, $a_{ll}$ and climatic niche are subject to $\alpha$. The climatic niche is assumed to evolve towards local conditions, resulting in a continuously more specialised adaptation to the local environment the longer a reference plant persists in a locality:

$$C_{leaf, new} = C_{leaf, old} + \mathcal{N}(\alpha, 0.05) \cdot (C_{leaf, target} - C_{leaf, old}) \quad (18)$$

$$a_{ll, new} = a_{ll, old} + \mathcal{N}(\alpha, 0.05) \cdot (a_{ll, target} - a_{ll, old}) \quad (19)$$

$$T_{min, plant, new} = T_{min, plant, old} + \mathcal{N}(\alpha, 0.05) \cdot (T_{coldestmonth, local} - T_{min, plant, old}) \quad (20)$$

$$T_{max, plant, new} = T_{max, plant, old} + \mathcal{N}(\alpha, 0.05) \cdot (T_{warmestmonth, local} - T_{max, plant, old}) \quad (21)$$

$$T_{mean, plant, new} = T_{mean, plant, old} + \mathcal{N}(\alpha, 0.05) \cdot (T_{mean, local} - T_{mean, plant, old}) \quad (22)$$

Where "target" denotes the trait value predicted to be optimal from the optimisation procedure, and "local" is the local climatic variable. To allow a limited degree stochasticity in the evolutionary adaptation in space and time $\mathcal{N}(\alpha, 0.05)$ indicates that the rate of evolution is drawn from a normal distribution with mean $\alpha$ as defined for the simulation run and a standard deviation of 0.05. The height trait is considered to be fully dynamic and set to the evaluated maximum potential height with a plant's current trait combination, accounting for the dynamic response of plant heights to environmental

conditions. However, given that $C_{leaf}$ is subject to $\alpha$, there is some degree of conservatism in the growth form between time steps. In locations where a deciduous phenology is assessed to result in a favourable NCG, and where $a_{ll,new}$ is ≤1 year, the modelled plant is considered deciduous.

In addition to trait adaptation, vegetation can respond to environmental changes through dispersal. At every time step, the modelled plants can disperse into the surrounding grid cells within a user-defined range. By testing variable dispersal capacities of the vegetation, the importance of trait adaptation vs dispersal and redistribution dynamics is tested. Dispersal is modelled as a stochastic process, accounting for the large range of factors involved in plant dispersal but not explicitly represented here (e.g., dispersal vectors, weather, soil properties)[79]. At every time step and for every modelled plant, a new dispersal distance is drawn from a dispersal kernel, defined by a Weibull distribution with a shape parameter value of 1.75 and a user-defined scale parameter. In this study, the dispersal scale is varied between simulations in the range from 200 to 1100 km per 10 kyr, covering a large range in possible dispersal capacities as inferred from plant fossils for the Paleocene–Eocene Thermal Maximum (PETM)[5]. The shape of the dispersal kernel accounts for the general right-skewed distribution of dispersal distances, with a high frequency of low dispersal distances and a low frequency of long-distance dispersals[67]. During dispersal, offspring with trait combinations identical to those of the reference plant migrate into the grid cells within the distance of the drawn dispersal value.

If several reference plants with a positive NCG occur in a location after dispersal, a competition function selects the surviving reference plant with the best trait combination. It is assumed that the reference plant with the highest NCG and plant height will dominate the local vegetation, considering carbon gain and success in light competition as the main fitness measures. To account for the combined height and NCG competition during the selection, NCG is scaled with a height penalty function that depends on the average height of the competing reference plants[34].

To model trait evolution, dispersal and competition dynamics, the described model functions are implemented in the "general engine for eco-evolutionary simulations" framework (gen3sis)[31]. The framework enables efficient handling of the described eco-evolutionary steps while considering a spatially-explicit dynamic landscape with temporally variable climatic input data.

The choice of the time stepping in the TREED model is variable and user-defined. In this study, a time step of 10 kyr was employed to resolve the abruptness of the PETM onset climate warming, while still being able to cover the full range of the PETM of 200 kyr, and to test a large range of dispersal, evolution, warming and $CO_2$ fertilisation scenarios. Thus, in the presented implementation, at every 10 kyr time step, the model takes multiyear monthly average climatologies as inputs and calculates average vegetation structures and carbon fluxes, but does not resolve variability at any other temporal resolution in between 10 kyr time steps and apart from the seasonal variability derived from the monthly climate inputs.

**Soil organic carbon.** Soil carbon stocks are based on a reduced-complexity soil carbon model derived from[35]. In this model, a one-dimensional soil layer at steady state is considered, where inputs from NPP are equal to heterotrophic respiration. It is assumed that soil heterotrophic respiration is directly proportional to the soil carbon pool and modulated by temperature according to a $Q_{10}$ function with a baseline temperature of 15 °C[80]:

$$C_{soil} = \frac{NPP}{k \cdot Q_{10}^{(T-15)/10}} \quad (23)$$

with a uniform $k$ of $\frac{1}{16}$ year$^{-1}$ and a $Q_{10}$ of 1.75.

## Climate data

The TREED vegetation model is forced with monthly output from climate simulations using the Community Earth System Model version 1.2 (CESM1.2)[6]. For the CESM1.2 climate simulations, a high-resolution fixed sea surface temperature configuration with coupling between the atmosphere component, CAM5.3 (Community Atmosphere Model version 5.3), the land component CLM4 (Community Land Model version 4), and the River Transport Model (RTM)[81–83] was applied. Sea surface temperatures were derived from lower-resolution (≈ 2°) fully-coupled companion simulations, run to equilibrium (≈ 1800 years). CESM1.2 model boundary conditions were taken from Deep-MIP forcing datasets[84,85]. Two climate simulations are used as the basis for the vegetation modelling, one with a pre-PETM atmospheric $CO_2$ concentration of 680 ppm and one with a PETM $CO_2$ of 1590 ppm. The simulations were carried out assuming a neutral orbital configuration (0 eccentricity, obliquity of 23.5°).

For the continuous TREED vegetation model simulations, climate conditions before the PETM onset are derived from the CESM1.2 pre-PETM simulation at 680 ppm atmospheric $CO_2$ and are kept constant during a 50 kyr period. After this period, the TREED model is forced with the PETM climate outputs (1590 ppm $CO_2$), with no intermediate climate conditions (step change in climate, corresponding to a PETM onset duration of 10 kyr). Then, assuming a PETM body duration of 70 kyr, $CO_2$ concentration and climate are kept constant at 1590 ppm $CO_2$. Finally, during the PETM recovery period, the $CO_2$ concentration is assumed to linearly decrease to 780 ppm $CO_2$ until 150 kyr after the PETM onset, and to pre-PETM conditions of 680 ppm $CO_2$ until 200 kyr after the PETM onset. For the transition climates, the approach from other deep-time biogeochemical models[86,87] is followed, and climatic fields are interpolated from the available climate model runs. The same procedure is applied for testing the effects of different land surface warming scenarios on vegetation dynamics (i.e., Fig. 3). The PETM $CO_2$ forcing and the resulting climatic fields are illustrated in Supplementary Fig. 7.

## Palynofloral trait reconstructions

The palynofloral records compiled in ref. 6 are used for the PETM vegetation trait reconstructions. Only records with a degree of age control that allows differentiation of "PETM body" and "PETM recovery period" samples are considered, in agreement with the carbon isotope records shown in Fig. 1. To derive vegetation traits from the fossil spore and pollen record, the nearest living relative (NLR) approach is used. Instead of assuming trait conservatism in the climatic ranges between fossil species and NLRs, as in the traditional NLR approach[6], we assume conservatism in plant traits. We consider the two approaches similar in their underlying assumption, given that plant traits ultimately determine the ability of species to persist in a given environment. In the trait estimation procedure, each fossil species is assigned an average trait value derived from observed trait values of their NLR under present-day conditions. The NLRs for the fossil species observed in the spore and pollen data are derived from ref. 6. Present-day trait values are derived from the communal plant trait database TRY[88], with a full reference list of the considered trait measurements given in the Supplementary References. The traits plant height, leaf mass per area (= 1/SLA), and phenology are considered. Regarding phenology observations in the TRY database, a value of 0 is assigned to observations of deciduousness, and 1 assigned to observations of an evergreen habit. From the available trait data, average values are calculated on the family and genus level and assigned to the fossil species of the same family or genus. With every observation in the TRY database being given the same weight, the average will approximate the most common trait value within a genus or family. While not representing the full range of variability observed in natural systems, or within a family or genus, the generalised approach aims to approximate the major ranking of trait values among the involved taxa (Supplementary Fig. 9). If a fossil species is assigned several possible nearest living relatives, an average value across all observations from the plausible genera and families is assigned. Genus-based trait assignments may give more specific trait values than family ones, due to a higher degree of trait conservatism at a lower taxonomic level. However, not all fossil species have living relatives at the genus level, resulting in a low coverage of trait values for the observed fossil species. This is why both genus-based and family-based reconstructions are presented. To give a temporal evolution of the average trait combination at the different localities, the assigned trait values are weighted by the abundance of the fossil species in the sample. An illustration of the full range of trait values in the different samples and their relative abundances is given in Supplementary Figs. 11, 12 and 13.

## Silicate weathering estimation

Silicate weathering rates throughout the PETM are estimated considering the CESM climate inputs, the topographic reconstruction of ref. 84 and vegetation biomass estimates from the TREED vegetation model. We apply a weathering equation commonly used in deep-time biogeochemical models[86,89]:

$$\omega_{silw} = \chi_m \cdot \epsilon \cdot \left(1 - \exp\left[-K \cdot e^{\frac{E_a}{RT_0} - \frac{E_a}{RT}} \cdot (1 - e^{-k_w \cdot q}) \cdot \frac{(z/\epsilon)^{\sigma+1}}{\sigma+1}\right]\right) \cdot f_{BM}$$

(24)

where $\omega_{silw}$ is the cation flux released during silicate weathering and $\epsilon$ is the local erosion, calculated as a function of topographic slope ($s$), runoff ($q$), temperature ($T$) and a scale parameter ($k_e$; calibrated to reproduce erosion rates similar to present-day levels) following[90]:

$$\epsilon = k_e \cdot q^{0.31} \cdot s \cdot \max(T, 2)$$

(25)

In Eq. (24), $T$ is the local temperature, $X_m$ the cation abundance in bedrock (0.1), $z$ the regolith thickness (10 m), $E_a$ the apparent activation energy of silicate weathering (20 KJ mol⁻¹), $R$ the ideal gas constant, and $T_0$ the standard temperature. $K$, $k_w$ and $\sigma$ are calibration constants ($6 \cdot 10^{-5}$, $10^{-3}$ and $-0.1$, respectively, adopted from ref. 86). Finally, $f_{BM}$ is a vegetation-dependent weathering enhancement factor. Based on the "carbon-energy flux" hypothesis[24], weathering enhancement is assumed to scale with carbon and nutrient fluxes, which intensify with biomass, resulting in a weathering enhancement formulation derived from ref. 36:

$$f_{BM} = [(1 - \min(BM_{norm}, 1) \cdot NOPLANT + BM_{norm}]$$

(26)

where $BM_{norm}$ is a normalised biomass indicator between 0 and 1. The biomass indicator linearly scales with the biomass density derived from the TREED model, with 0 indicating no biomass, and 1 indicating biomass densities as observed in today's tropical forests (140 Mg C per ha, or around 300 Mg dry biomass per ha)[91]. NOPLANT describes the background weathering intensity in the absence of vegetation. In agreement with laboratory and field studies, a maximum six-fold weathering enhancement by vegetation is assumed (NOPLANT = 1/6)[22,23,25]. The $CO_2$ consumption by silicate weathering ($F_{silw}$) is assumed to be proportional to the cation release $\omega_{silw}$[86], with $1 \cdot 10^{13}$ mol C year⁻¹ as the baseline carbon consumption ($k_{silw}$; in the range of present-day carbon consumption rates through silicate weathering) and $\omega_{ref}$ as the baseline cation release flux under the initial climatic conditions:

$$F_{silw} = k_{silw} \cdot \frac{\omega_{silw}}{\omega_{ref}}$$

(27)

## Data availability

All data processed and generated in this study are available on Zenodo under https://doi.org/10.5281/zenodo.17252973.

## Code availability

All computer code necessary to reproduce the presented results, including the vegetation model code and scripts for analysing the model results and the data, are available on Zenodo under https://doi.org/10.5281/zenodo.17252973. The figures and maps presented in the paper and Supplementary Information were produced using the open-source software R[92], visualisation package ggplot2[93], and using the packages raster[94], terra[95] and sf[96] for processing spatial data, with all scripts and data provided in the Zenodo data and code repository.

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

## Acknowledgements

This work was funded by the Swiss National Science Foundation grants 200021_192296 (to J.R. and T.V.G.), and 200021_231594 (to T.V.G.), the Australian Research Council Discovery Early Career Researcher Award grant 109938 (to V.A.K) and the U.S. National Science Foundation grant EAR-2121170 (to G.J.B). C.A.S. thanks the Heising-Simons Foundation (HSF) for supporting the climate modelling research, as well as the Computational and Information Systems Laboratory System NCAR Strategic Capability (NSC), Boulder, CO: NSF National Centre for Atmospheric Research, and the Extreme Science and Engineering Discovery Environment (XSEDE), which is supported by National Science Foundation grant ACI-1548562, as well as Stampede2 at the Texas Advanced Computing Centre, University of Texas at Austinthrough allocation TG-ATM180001. We thank Ellen Currano for making the LMA estimates from the Bighorn Basin available. We thank Melissa Dawes for English editing. We thank Scott Wing from the Smithsonian National Museum of Natural History in Washington for discussions on earlier versions of this manuscript. All computations except for the CESM climate simulations were performed on the ETH Zurich "Euler" computer cluster.

## Author contributions

J.R., L.P. and T.V.G. developed the concept and designed the study. J.R. developed the vegetation model, conducted the modelling and analysis, the palynofloral trait reconstructions, made the figures and the original draft of the manuscript. V.A.K. contributed to the palynofloral reconstruction and interpretation. G.J.B. advised on the geochemical context of the study and provided data. C.A.S. conducted the climate modelling. J.R., V.A.K., G.J.B., C.A.S., T.V.G. and L.P. contributed to the interpretation of the results and the writing of the manuscript.

## Competing interests

The authors declare no competing interests.
