## [Transparent Peer Review file · Nature Communications]

Loss of vegetation functions during the Paleocene-Eocene Thermal Maximum

Corresponding Author: Dr Julian Rogger

Version 0:

Reviewer comments:

Reviewer #1

(Remarks to the Author)

Rogger et al. present results from a new eco-evolutionary vegetation model (TREED), driven by boundary conditions from climate model simulations using the Community Earth System Model of the pre-PETM and PETM, in order to constrain the vegetation response across this past global warming event. The authors evaluate the model against proxy reconstruction of vegetation response at three sites (two in the Northern mid-latitudes and one in the Arctic) based on pollen assemblages and using nearest living relatives to reconstruct modeled vegetation traits. The experimental framework evaluates the relative role of adaptation speed and dispersal as controls on the vegetation response and contrasts cases with and without CO₂ fertilization and different degrees of warming. Based on the response of vegetation, the model further constrains the impact on terrestrial carbon storage (by combining the vegetation model with a simple representation of soil carbon storage) and the impact on silicate weathering (by adding a scaling of the weathering function based on biomass density). Combined, the model sensitivity tests and comparison with vegetation reconstructions from the three sites suggest that PETM warming exceeded a threshold for the ability of vegetation to adapt or migrate, thus leading to a temporary loss in productivity and hence total land carbon storage as well as a counteracting influence on the warming-induced increase in silicate weathering. The modeled vegetation response is consistent with both a long-duration of the so-called 'PETM body' and abrupt recovery.

I think this is a great study and very useful modeling contribution, with relevance both for the PETM and for the long-term future response of vegetation to anthropogenic warming. The manuscript is well written, the model description is thorough and the presentation of results is easy to follow, both in the text and the figures. At the same time, there is a lot included in the manuscript, so I have a few suggestions, which may help contextualize the results. Overall, I think this paper is really well done and worthy of publication in Nature Communications.

First, in a few places in the introduction (lines 46, 52, and 56) the text doesn't clearly discriminate between biotic carbon stocks versus terrestrial carbon stocks specifically. Slight changes to the wording could clarify that existing $\delta^{13}\text{C}$ records do not uniquely implicate terrestrial organic carbon as a capacitor, as opposed to other organic carbon sources (marine organic carbon, hydrates, permafrost carbon, etc.). Changes in terrestrial organic carbon storage (vegetation and shallow soils, excluding permafrost) are one possibility, and this paper is evaluating the hypothesis that the terrestrial ecosystem response can account for the highlighted aspects of the global $\delta^{13}\text{C}$ records.

Next, the Methods section answered most of the model-specific questions that I had while reading the text, but I think it would be helpful to include descriptions of the following in the main text ('Trait based vegetation model' section) in a third paragraph before Results: 1) a brief description of how the vegetation model interacts with soils - e.g. the sentence in the Figure 1 caption that soil carbon is a function of net primary productivity and temperature-dependent heterotrophic respiration, 2) a brief description of how silicate weathering is calculated - e.g. move forward the description on Line 160 and clarification of whether soils and weathering are part of the vegetation model or computed offline, 3) a brief description of the experimental framework, e.g. how the time-dependent changes were constrained (50 kyr using pre-PETM constraints, 70 kyr of PETM constraints, then a linear CO₂ decrease over 130 kyr with linear interpolation of climatic fields), and 4) a brief summary of the sensitivity tests - varying the adaptation and dispersal capacity and including CO₂ fertilization and different degrees of warming. Specifically, how are different amounts of warming imposed? What is the amount of warming that is consistent with the CESM simulations (pre-PETM to PETM)? Is scaling applied to both spatial patterns in temperature change as well as the seasonal temperature range? These additional descriptions of both the model and the experimental

framework would allow much more clarity in interpreting the subsequently presented results.

A few more specific model questions (but that could go into the Methods):

Line 99: are modern orbits assumed in calculating seasonality in both the pre-PETM and PETM simulations with CESM?

Line 523: what is the rationale behind selecting a 10 kyr model timestep?

Line 572: what is the temporal variation of climate input data? Monthly?

In the discussion, it would be helpful to add some text to clarify the lack of feedback between the imposed climate boundary conditions and vegetation response and consider implications of that assumption. For instance, a modeled drop in biomass should generate not only a CO₂ feedback via carbon release but also have a climate influence via changes in albedo, evapotranspiration, etc., and these cannot be represented with this experimental framework. Such feedback could potentially influence the timescale of the vegetation-mediated climate response.

Further, how have the results been shaped by assumptions in the modeling framework - specifically, the imposition of the PETM climate for 70 kyr? What if, instead, the experiment imposed a linear recovery in CO₂ and temperature without a lag and more closely approximating the modeled response to a single pulse release of CO₂ as has been simulated in models of the ocean-atmosphere (e.g. cGENIE, LOSCAR)? Would there still be a prolonged reduction in productivity, which in turn might explain why the actual PETM warmth persisted over a long duration? Given the number of sensitivity tests already included in the study - evaluating trait adaptation capacity, dispersal capacity, CO₂ fertilization and degree of warming - I am not suggesting that it would necessarily add clarity to evaluate different assumptions about the climate boundary conditions, but it would be useful to discuss how the assumptions about these boundary conditions have impacted the results.

Finally, in making comparisons between results for the PETM and the long-term future, do the authors have a sense whether the response is related to the relative or total amount of warming? Is 4 deg C a threshold because of an absolute temperature reached, or would 4 degrees of warming from a lower baseline have similar results, particularly because of the climatic niche constraint?

(Remarks on code availability)

I did not try to run the model but I reviewed the files the authors have provided online, particularly the README.

Reviewer #2

(Remarks to the Author)

Review of: "Loss of vegetation functions during the Paleocene-Eocene thermal maximum" by Rogger, J., Korasidis, V. A., Bowen, G. J., Shields, C. A., Gerya, T. V., and Pellissier, L., manuscript submitted to Nature Communications

I read with much interest the manuscript by Rogger et al. It is a quite interesting study with significant novelty that deserves to be published in Nature Communications.

The authors present a new vegetation model with the originality that it is not based on pre-defined plant functional types, but on the optimization of plant traits to the local climate. This offers the advantage of plant traits varying in a continuous way in the modelled vegetation. Furthermore, the model can take into account adaptation lags under changing climate, as well as dispersal from nearby pixels. This is a rather unique tool to study the response of natural vegetation to palaeoclimate changes.

The model is tested over the period of Paleocene-Eocene Thermal Maximum (PETM). The PETM warming was likely too fast and too intense to allow the adaptation of the terrestrial vegetation. For global land surface warmings larger than 4°C, the model predicts a strong drop of net primary productivity and biomass of the terrestrial biosphere, that presumably impacted the carbon cycle and silicate weathering, implying a positive feedback on atmospheric CO₂. This is an important result, which may have implications in the framework of the present climate warming.

The current version of the paper is already a revised version. I had not the opportunity to comment on the original version. Nevertheless, I checked that the authors responded completely to all reviewers' comments. Their response looks fine to me. One of the main points raised was linked to the reconstructions of palaeo plant traits from palynofloral records. Indeed, plant traits are reconstructed at the genus or family level (which is the best that can be done from palynoflora) and assumes that the average genus or family traits can be obtained from the modern TRY database, i.e., that these average traits at genus or family levels have not changed from the Paleocene-Eocene to the present. This may look inconsistent with the idea (and the model) that traits optimize in response to local climate. It is well-known that traits can vary strongly between individuals within a genus, or even within a single species. However, the current knowledge does not allow to make precise reconstruction of past trait values through times, especially for the PETM where macrofloral data may not provide the necessary resolution to assess the evolution through the PETM. Anyway, the authors now reconstruct LMA trait from macrofloral data at one of their sites (Bighorn site) where such a reconstruction is possible. The trends are consistent with the results obtained from the palynoflora. But, more importantly, the authors now state clearly their hypothesis, especially that the aim is not to reconstruct absolute values of traits (which is probably not possible), but relative changes over the PETM period. The hypothesis is then

that the relative differences between genera or families have not changed since the Paleocene. This is not necessarily true, but it is fine provided that the hypothesis is clearly stated.

I have noted a few other problems that I am summarizing below. The paper can be accepted after these small changes are taken into account.

The most significant problem is related to the validation of the model on the present (Supplementary figure 2). This is important, since the model is new and rests on a slightly different methodology compared to classical dynamic vegetation models. The authors just provide maps of height, NPP, AGB and AET that they compare visually with similar maps obtained from upscaled data or remote sensing to validate the model. This is not enough, especially that significant differences can be observed with the data. They should report statistical measures, such as correlation coefficients, RMSE and biases between modelled and observed maps.

The other point is that they are not validating model traits on present data (except height), which is the main outputs they use from the model in their study. They should compute at least at the level of a few sites (or regions) average traits values for LMA and deciduousness, from the genera or families present at these sites, using values from TRY, exactly as they do for the PETM.

Some information should be added for several equations in the methodology section. For instance, equation (1) is not unit-independent. It is necessary to provide the units for SLA and all. Similarly, the units of the conductances should be provided in equation (9) ($g_m = 3.26$ is a conductance and should have a unit). Also, in equations (18) to (22), the function $N(\alpha, 0.05)$ should be clearly defined.

In equation (55), canopy conductance responds to (soil) water availability, but not to air relative humidity or saturation deficit, as in usual formulations of canopy conductance. This may need some comments.

It looks that the model neglects the diurnal cycle of photosynthesis and uses monthly average climatic data to calculate its rate. This is a strong approximation, that might be justified for computation efficiency. However, the impact on the absolute value of photosynthesis may be very important, because the process is not linear. Anyway, since relative differences are more important in this study, than absolute values, this hypothesis is probably fine, but it should be discussed.

In equation (57), R_n is daytime net radiation. It would be useful to provide some explanation on the way it is evaluated. It is just interpolated from the climate model outputs or does it react to the vegetation changes (e.g., through LAI impact on albedo) calculated by the vegetation model ?

P 63, l 1282: km instead of km⁻¹

(Remarks on code availability)

Reviewer #3

(Remarks to the Author)

I thank the authors for this interesting and well-presented manuscript. The study is well designed, thorough and encompasses novel data compilation and modelling. The results are discussed coherently and extensively. Several key results stand out that are of high relevance to researchers in the fields of paleo and modern climate change and could have policy implications. To me, the key results are: a threshold warming beyond which terrestrial productivity is strongly reduced; vegetation changes during the PETM differed by latitude and productivity might have been stable at high latitudes; vegetation changes during the PETM would have contributed to the recorded $\delta^{13}C$ change across the event and could have controlled the strength and time scale of the silicate weathering feedback. These are all important findings that merit publication in a high profile journal. I suggest accepting it with minor revisions. It is a fascinating read and relevant to a wide audience.

My main comments are the following:

1) I think this study contains multiple important results which could be spelled out more explicitly and developed further:

A - the threshold for exceeding dispersal strategy

The existence of a threshold warming for the capacity of vegetation to cope with warming is an important result in my view and of interest for the understanding of paleo events and the human-made climate change. How does this threshold depend on the speed of climate change? Rather than as an absolute temperature change, could this threshold be expressed as a warming rate?

B - implications for the interpretation of the $\delta^{13}C$ anomalies during the PETM

As the authors mention multiple times, $\delta^{13}C$ changes in marine carbonates are a key feature of the geologic record of the PETM. It has even been used to estimate the magnitude of carbon fluxes during the event, usually without considering

terrestrial d13C fluxes (e.g. Gutjahr et al. 2017, Komar and Zeebe, 2017, Haynes et al. 2020). The authors already allude to it but I think it would be interesting and could increase the relevance and reach of this paper by adding a brief discussion of the implications of the simulated terrestrial carbon stock changes in the present study would affect these estimates, and to put the simulated carbon fluxes in relation to those previously estimated.

2) I think it would be a valuable addition to discuss how these results depend on the background state, e.g. how applicable they are to other climate events. The PETM starts from a much warmer background climate than today, with higher atmospheric CO₂. How does this affect the potential of plants to adapt or migrate under warming compared to today? Does the CO₂ fertilization effect tail off when atmospheric CO₂ is above a certain threshold?

3) Does the vegetation model consider disturbances like fire, storm or pest damage, which might have increased due to the PETM warming? If not, how might including these processes alter the results? Do they represent a best-case scenario?

References

Gutjahr, M., Ridgwell, A., Sexton, P.F., Anagnostou, E., Pearson, P.N., Pälike, H., Norris, R.D., Thomas, E. and Foster, G.L., 2017. Very large release of mostly volcanic carbon during the Palaeocene–Eocene Thermal Maximum. *Nature*, 548(7669), pp.573-577.

Haynes, L.L. and Hönisch, B., 2020. The seawater carbon inventory at the Paleocene–Eocene Thermal Maximum. *Proceedings of the National Academy of Sciences*, 117(39), pp.24088-24095.

Komar, N. and Zeebe, R.E., 2017. Redox-controlled carbon and phosphorus burial: A mechanism for enhanced organic carbon sequestration during the PETM. *Earth and Planetary Science Letters*, 479, pp.71-82.

(Remarks on code availability)

The code is very well documented and seems complete. I didn't test all scripts but the one I tested worked fine.

Version 1:

Reviewer comments:

Reviewer #1

(Remarks to the Author)

I reviewed a previous version of this manuscript. The authors have addressed all of my comments in both their response and revision. I think this is an excellent study and worthy of publication in its present form.

(Remarks on code availability)

Reviewer #2

(Remarks to the Author)

Review of: "Loss of vegetation functions during the Paleocene-Eocene thermal maximum" by Rogger, J., Korasidis, V. A., Bowen, G. J., Shields, C. A., Gerya, T. V., and Pellissier, L., manuscript submitted to *Nature Communications*

I read carefully the revised version of the manuscript. The authors addressed all the comments I made in my first review and modified the main manuscript and the supplementary text accordingly. From my point of view, the points raised by the other reviewers were also quite fully addressed. As a result, the paper can be published in its current form, after corrections of a few typos/grammatical errors I found while reading:

Page 16, line 369: "a carbon cycle" instead of "a carbon carbon cycle"

Page 17, line 411: "by around an order of magnitude" instead of "by around a magnitude"

Page 20, line 462: "are used to describe" instead of "are used describe"

Page 48, Legend of Supplementary Figure 6, A & B: the figures show LMA versus latitude, while the legend refers to SLA. All occurrences of SLA in the legend should be replaced by LMA. Also, legend for figure D is missing.

(Remarks on code availability)

Reviewer #3

(Remarks to the Author)

I thank the author team for their efforts in revising the manuscript.

I re-read the revised manuscript a few times now and think that, while future modelling studies should take more of the

complexity of the coupled atmosphere-land-ocean system into account, this current manuscript presents a well rounded study and will surely spark important debates. I look forward to seeing this manuscript published.

(Remarks on code availability)

I could not access the source code via the provided link

Reviewer #1:

Rogger et al. present results from a new eco-evolutionary vegetation model (TREED), driven by boundary conditions from climate model simulations using the Community Earth System Model of the pre-PETM and PETM, in order to constrain the vegetation response across this past global warming event. The authors evaluate the model against proxy reconstruction of vegetation response at three sites (two in the Northern mid-latitudes and one in the Arctic) based on pollen assemblages and using nearest living relatives to reconstruct modeled vegetation traits. The experimental framework evaluates the relative role of adaptation speed and dispersal as controls on the vegetation response and contrasts cases with and without CO₂ fertilization and different degrees of warming. Based on the response of vegetation, the model further constrains the impact on terrestrial carbon storage (by combining the vegetation model with a simple representation of soil carbon storage) and the impact on silicate weathering (by adding a scaling of the weathering function based on biomass density). Combined, the model sensitivity tests and comparison with vegetation reconstructions from the three sites suggest that PETM warming exceeded a threshold for the ability of vegetation to adapt or migrate, thus leading to a temporary loss in productivity and hence total land carbon storage as well as a counteracting influence on the warming-induced increase in silicate weathering. The modeled vegetation response is consistent with both a long-duration of the so-called 'PETM body' and abrupt recovery.

I think this is a great study and very useful modeling contribution, with relevance both for the PETM and for the long-term future response of vegetation to anthropogenic warming. The manuscript is well written, the model description is thorough and the presentation of results is easy to follow, both in the text and the figures. At the same time, there is a lot included in the manuscript, so I have a few suggestions, which may help contextualize the results. Overall, I think this paper is really well done and worthy of publication in Nature Communications.

We thank the reviewer for taking the time to review our manuscript, the positive feedback and the useful suggestions which have helped to improve the clarity of the manuscript and its conclusions.

First, in a few places in the introduction (lines 46, 52, and 56) the text doesn't clearly discriminate between biotic carbon stocks versus terrestrial carbon stocks specifically. Slight changes to the wording could clarify that existing d¹³C records do not uniquely implicate terrestrial organic carbon as a capacitor, as opposed to other organic carbon sources (marine organic carbon, hydrates, permafrost carbon, etc.). Changes in terrestrial organic carbon storage (vegetation and shallow soils, excluding permafrost) are one possibility, and this paper is evaluating the hypothesis that the terrestrial ecosystem response can account for the highlighted aspects of the global d¹³C records.

We thank the reviewer for ensuring clarity and we have revised the text to avoid implying that vegetation and soil carbon stocks are the only capacitor system of potential importance during the PETM. Particularly, we now mention other suggested capacitor systems (e.g., methane hydrates, permafrost) and clarified that we focus on the role of vegetation and soil carbon stocks (L43-50).

Next, the Methods section answered most of the model-specific questions that I had while reading the text, but I think it would be helpful to include descriptions of the following in the main text ('Trait based vegetation model' section) in a third paragraph before Results:

- 1) a brief description of how the vegetation model interacts with soils - e.g. the sentence in the Figure 1 caption that soil carbon is a function of net primary productivity and temperature-

dependent heterotrophic respiration, 2) a brief description of how silicate weathering is calculated - e.g. move forward the description on Line 160 and clarification of whether soils and weathering are part of the vegetation model or computed offline, 3) a brief description of the experimental framework, e.g. how the time-dependent changes were constrained (50 kyr using pre-PETM constraints, 70 kyr of PETM constraints, then a linear CO₂ decrease over 130 kyr with linear interpolation of climatic fields), and 4) a brief summary of the sensitivity tests - varying the adaptation and dispersal capacity and including CO₂ fertilization and different degrees of warming. Specifically, how are different amounts of warming imposed? What is the amount of warming that is consistent with the CESM simulations (pre-PETM to PETM)? Is scaling applied to both spatial patterns in temperature change as well as the seasonal temperature range? These additional descriptions of both the model and the experimental framework would allow much more clarity in interpreting the subsequently presented results.

We have extended the methodological description in the main part of the manuscript as suggested by the reviewer (L103-145). We have clarified that soil carbon and silicate weathering are computed offline based on the results of the vegetation model and include details on their calculation in the main text. Further, we have added a third paragraph to the model description describing the climate boundary conditions used for the vegetation modelling, that the warming scenarios were derived by assuming different maximum CO₂ concentrations during the PETM body, that the reference warming scenario from the original CESM simulations results in a 8°C land surface warming, and that monthly climate fields required by the vegetation model were derived by linear interpolation to the specified CO₂ level from the available CESM simulations.

A few more specific model questions (but that could go into the Methods):

Line 99: are modern orbits assumed in calculating seasonality in both the pre-PETM and PETM simulations with CESM?

The CESM simulations used in this study were conducted considering a neutral orbital configuration with 0 eccentricity and 23.5° obliquity, which we have now clarified on L632-633.

Line 523: what is the rationale behind selecting a 10 kyr model timestep?

We have clarified on L608-615 that the choice of the modelling time step in the vegetation model is arbitrary. Here, 10 kyr was chosen to have a reasonably abrupt onset of the PETM warming, while still being able to computationally cover the full 200 kyr length of the PETM and to conduct a large range of sensitivity tests regarding rate of evolution, dispersal, magnitude of warming and CO₂ fertilization.

Line 572: what is the temporal variation of climate input data? Monthly?

We have clarified in the main manuscript and the Methods section that the model uses monthly climate fields at each timestep (L135-136; L611-615)

In the discussion, it would be helpful to add some text to clarify the lack of feedback between the imposed climate boundary conditions and vegetation response and consider implications of that assumption. For instance, a modeled drop in biomass should generate not only a CO₂ feedback via carbon release but also have a climate influence via changes in albedo, evapotranspiration, etc., and these cannot be represented with this experimental framework. Such feedback could potentially influence the timescale of the vegetation-mediated climate response.

We have clarified in the discussion (L389-399) that by imposing climatic boundary conditions, we do not consider such feedback with the climate system or the geological carbon cycle. We have further added to the discussion that the consideration of such interactions in a fully coupled manner will be important to understand how a potential productivity loss – as suggested by our results – may affect the magnitude and timescale of a PETM-like climate perturbation through positive feedback dynamics (e.g., release of biomass carbon).

Further, how have the results been shaped by assumptions in the modeling framework - specifically, the imposition of the PETM climate for 70 kyr? What if, instead, the experiment imposed a linear recovery in CO₂ and temperature without a lag and more closely approximating the modeled response to a single pulse release of CO₂ as has been simulated in models of the ocean-atmosphere (e.g. cGENIE, LOSCAR)? Would there still be a prolonged reduction in productivity, which in turn might explain why the actual PETM warmth persisted over a long duration? Given the number of sensitivity tests already included in the study - evaluating trait adaptation capacity, dispersal capacity, CO₂ fertilization and degree of warming - I am not suggesting that it would necessarily add clarity to evaluate different assumptions about the climate boundary conditions, but it would be useful to discuss how the assumptions about these boundary conditions have impacted the results.

We thank the reviewer for highlighting this important discussion point, which we believe is key to communicate to the reader. Indeed, carbon cycle models like cGENIE and LOSCAR which focus strongly on ocean biogeochemistry generally model a fast onset of the climatic recovery after an initial pulse of carbon input. To obtain a prolonged warming as observed in proxy records (e.g., Fig 1A) they require additional carbon inputs during the PETM body (e.g., Penman et al. 2016, Zeebe et al. 2009). In these models, it is assumed that silicate weathering on land and subsequent carbonate sedimentation in the ocean effectively regulate climate as soon as temperatures rise. If we imposed a faster recovery of climate in our model, vegetation would recover faster as well, but there would still be a period of disrupted climate regulation for as long as the warming magnitude exceeds the adaptation capacity through dispersal and evolution. This temporary loss could then trigger positive feedback dynamics, including a loss of biospheric carbon storage and weathering, further prolonging the warming. We state in the discussion that the delay in weathering and organic carbon sequestration as a result of a vegetation disruption may offer an additional explanation for the delay in the climate regulation observed during the PETM (L365-385) and we now more clearly highlight (L389-399) that it may well be the response of vegetation that could determine the magnitude and duration of the PETM warming, as the reviewer suggests. To evaluate the effects of the vegetation disruption on the timescale of the climatic recovery through positive feedback dynamics, the proposed adaptation processes should be considered in a fully coupled manner with the climate system and the geologic carbon cycle. The main goal of the presented work and with the developed model was to evaluate the possibility that there was a widespread loss of vegetation functions during the PETM. Integrating these dynamics in a fully coupled Earth system and carbon cycle model over geologic timescales is an endeavour we hope to support with our findings, but that will require considerable further research and model development.

Penman, D. E. et al. An abyssal carbonate compensation depth overshoot in the aftermath of the Palaeocene–Eocene Thermal Maximum. *Nature Geoscience* 9, 575–580 (2016).

Zeebe, R. E., Zachos, J. C. & Dickens, G. R. Carbon dioxide forcing alone insufficient to explain Palaeocene–Eocene Thermal Maximum warming. *Nature Geoscience* 2, 576–580 (2009).

Finally, in making comparisons between results for the PETM and the long-term future, do the authors have a sense whether the response is related to the relative or total amount of warming? Is 4 deg C a threshold because of an absolute temperature reached, or would 4

degrees of warming from a lower baseline have similar results, particularly because of the climatic niche constraint?

We have extended the discussion on the comparison between the PETM and anthropogenic climate change in line with the reviewer's comments (L400-420). The vegetation response will likely depend on both the total and relative amount of warming. We now state that the absolute temperature limit before a loss of productivity occurs may lay higher for the present because we start in a colder background climate state. Consequently, the high latitude positive effects of warming, creating environments more optimal for photosynthesis (warmer, increased CO₂) as we observed for the PETM, might extent into lower latitudes. However, the relative warming and the speed of warming are important, with current warming being considerably faster than during the PETM. This means less time for both dispersal and evolutionary adaptation. While both for the PETM and the present, slow evolution may induce an adaptive lag for a given relative climate shift, today, limited dispersal may have the potential to induce an additional adaptive lag because of the rapid speed of the climatic change.

Reviewer #1 (Remarks on code availability):

I did not try to run the model but I reviewed the files the authors have provided online, particularly the README.

Reviewer #2:

Review of: "Loss of vegetation functions during the Paleocene-Eocene thermal maximum" by Rogger, J., Korasidis, V. A., Bowen, G. J., Shields, C. A., Gerya, T. V., and Pellissier, L., manuscript submitted to Nature Communications

I read with much interest the manuscript by Rogger et al. It is a quite interesting study with significant novelty that deserves to be published in Nature Communications.

The authors present a new vegetation model with the originality that it is not based on pre-defined plant functional types, but on the optimization of plant traits to the local climate. This offers the advantage of plant traits varying in a continuous way in the modelled vegetation. Furthermore, the model can take into account adaptation lags under changing climate, as well as dispersal from nearby pixels. This is a rather unique tool to study the response of natural vegetation to palaeoclimate changes.

The model is tested over the period of Paleocene-Eocene Thermal Maximum (PETM). The PETM warming was likely too fast and too intense to allow the adaptation of the terrestrial vegetation. For global land surface warmings larger than 4°C, the model predicts a strong drop of net primary productivity and biomass of the terrestrial biosphere, that presumably impacted the carbon cycle and silicate weathering, implying a positive feedback on atmospheric CO₂. This is an important result, which may have implications in the framework of the present climate warming.

The current version of the paper is already a revised version. I had not the opportunity to comment on the original version. Nevertheless, I checked that the authors responded completely to all reviewers' comments. Their response looks fine to me. One of the main points raised was linked to the reconstructions of palaeo plant traits from palynofloral records. Indeed, plant traits are reconstructed at the genus or family level (which is the best that can be done from palynoflora) and assumes that the average genus or family traits can be obtained from the modern TRY database, i.e., that these average traits at genus or family levels have not changed from the Paleocene-Eocene to the present. This may look inconsistent with the idea (and the model) that traits optimize in response to local climate. It is well-know that traits can

vary strongly between individuals within a genus, or even within a single species. However, the current knowledge does not allow to make precise reconstruction of past trait values through times, especially for the PETM where macrofloral data may not provide the necessary resolution to assess the evolution through the PETM. Anyway, the authors now reconstruct LMA trait from macrofloral data at one of their sites (Bighorn site) where such a reconstruction is possible. The trends are consistent with the results obtained from the palynoflora. But, more importantly, the authors now state clearly their hypothesis, especially that the aim is not to reconstruct absolute values of traits (which is probably not possible), but relative changes over the PETM period. The hypothesis is then that the relative differences between genera or families have not changed since the Paleocene. This is not necessarily true, but it is fine provided that the hypothesis is clearly stated.

We thank the reviewer for the positive feedback and for taking the time to go through the previous revision comments as well as for providing valuable feedback to ensure clarity of our methodological approach.

I have noted a few other problems that I am summarizing below. The paper can be accepted after these small changes are taken into account.

The most significant problem is related to the validation of the model on the present (Supplementary figure 2). This is important, since the model is new and rests on a slightly different methodology compared to classical dynamic vegetation models. The authors just provide maps of height, NPP, AGB and AET that they compare visually with similar maps obtained from upscaled data or remote sensing to validate the model. This is not enough, especially that significant differences can be observed with the data. They should report statistical measures, such as correlation coefficients, RMSE and biases between modelled and observed maps.

Following the reviewer's suggestion, we have split the previous Supplementary Figure 2 into four figures (Supplementary Figs. 2-5, referred to on L103), showing the visual comparison between modelled and observed values, a plot showing the spatial bias, and a scatterplot showing the modelled vs. observed values for height, NPP, AGB and AET, including the RMSE and correlation coefficient.

The other point is that they are not validating model traits on present data (except height), which is the main outputs they use from the model in their study. They should compute at least at the level of a few sites (or regions) average traits values for LMA and deciduousness, from the genera or families present at these sites, using values from TRY, exactly as they do for the PETM.

We have included an additional validation plot, Supplementary Figure 6 (referred to on L103), for a global LMA and deciduousness validation, where we compare the modelled latitudinal gradient in LMA for evergreen and deciduous plants compared to trait observations, as well as the current distribution of deciduous trees to the model predictions. Regarding LMA, we show that the model captures the main feature of an opposing LMA gradient for evergreens and deciduous trees with the cooling of climate towards higher latitudes as observed in field data. Regarding deciduousness, the model broadly covers the locations of Europe, Eastern North America and East Asia, where deciduousness is an abundant plant strategy (deciduous trees composing more than 50% of the vegetation composition).

Some information should be added for several equations in the methodology section. For instance, equation (1) is not unit-independent. It is necessary to provide the units for SLA and all. Similarly, the units of the conductances should be provided in equation (9) ($g_m = 3.26$ is a

conductance and should have a unit). Also, in equations (18) to (22), the function $N(\alpha, 0.05)$ should be clearly defined.

We have revised Supplementary Table 1 and have included an additional Supplementary Table 2 to list all model parameters, constants and variables including their units (referred to on L451-452), and have clarified equations 18-22 (L572-575).

In equation (55), canopy conductance responds to (soil) water availability, but not to air relative humidity or saturation deficit, as in usual formulations of canopy conductance. This may need some comments.

We thank the reviewer for pointing out this ambiguity in the methodological description. In the revised methodological description (L1195-1215) we have clarified that the canopy conductance does depend on both the local water availability as well as the atmospheric water demand. However, for the atmospheric water demand we follow the LPJ vegetation model (Schaphoff et al. 2018, Sitch et al. 2003), by applying Monteith's empirical relation between evaporation efficiency and surface conductance and calculate the atmospheric demand using an equilibrium evapotranspiration rate as a function of temperature and radiation (Eq. 57). If the atmospheric demand exceeds the local water availability, canopy conductance and the canopy-atmosphere gas exchange parameter λ are recalculated in a bisection algorithm to find a conductance and photosynthetic rate that results in a closed water balance.

Schaphoff, S. et al. LPJmL4 – a dynamic global vegetation model with managed land – Part 1: Model description. *Geoscientific Model Development* 11, 1343–1375 (2018)

Sitch, S. et al. Evaluation of ecosystem dynamics, plant geography and terrestrial carbon cycling in the LPJ dynamic global vegetation model. *Global Change Biology* 9, 161–185 (2003).

It looks that the model neglects the diurnal cycle of photosynthesis and uses monthly average climatic data to calculate its rate. This is a strong approximation, that might be justified for computation efficiency. However, the impact on the absolute value of photosynthesis may be very important, because the process is not linear. Anyway, since relative differences are more important in this study, than absolute values, this hypothesis is probably fine, but it should be discussed.

We have clarified that the vegetation model uses monthly average climate inputs in the main manuscript (L135-136). Further, we clarified in the Methods (L608-615) that the vegetation model is developed to obtain average vegetation dynamics across long, geologic timescales. To cover such timescales, the model considers multi-year monthly average climate inputs at every 10 kyr timestep for which it is run but does not cover any variability at a higher temporal resolution.

In equation (57), R_n is daytime net radiation. It would be useful to provide some explanation on the way it is evaluated. Is it just interpolated from the climate model outputs or does it react to the vegetation changes (e.g., through LAI impact on albedo) calculated by the vegetation model?

In the discussion, we have clarified that with the current model implementation there are no interactions between the vegetation model and the climate system (L391-395) and further clarified on L1209 that R_n is derived from the climate inputs.

P 63, l 1282: km instead of km⁻¹

Changed accordingly, thank you (now L1338).

Reviewer #3 (Remarks to the Author):

I thank the authors for this interesting and well-presented manuscript. The study is well designed, thorough and encompasses novel data compilation and modelling. The results are discussed coherently and extensively. Several key results stand out that are of high relevance to researchers in the fields of paleo and modern climate change and could have policy implications. To me, the key results are: a threshold warming beyond which terrestrial productivity is strongly reduced; vegetation changes during the PETM differed by latitude and productivity might have been stable at high latitudes; vegetation changes during the PETM would have contributed to the recorded $\delta^{13}\text{C}$ change across the event and could have controlled the strength and time scale of the silicate weathering feedback. These are all important findings that merit publication in a high profile journal. I suggest accepting it with minor revisions. It is a fascinating read and relevant to a wide audience.

We thank the reviewer for taking the time to evaluate our manuscript, the positive feedback and the helpful comments to clearly communicate the implications of our findings to a broad audience.

My main comments are the following:

1) I think this study contains multiple important results which could be spelled out more explicitly and developed further:

A - the threshold for exceeding dispersal strategy

The existence of a threshold warming for the capacity of vegetation to cope with warming is an important result in my view and of interest for the understanding of paleo events and the human-made climate change. How does this threshold depend on the speed of climate change? Rather than as an absolute temperature change, could this threshold be expressed as a warming rate?

We have extended the discussion on how the vegetation response depends on the absolute warming and how the difference in the rate of warming between the PETM and the present may affect the occurrence of an adaptive lag (L406-420). A direct comparison of warming rates between the PETM and current climate change is challenging, given the large uncertainty in the duration of the PETM onset, with estimates ranging from a few thousand years to 20 kyr (Turner, 2018). On the investigated warming timescales of 1-8°C over 10 kyr (now stated on L412-413), we show that dispersal was likely not a major limiting factor, and a species redistribution may have been able to compensate warming of up to 4°C land surface warming, after which climate adaptation through evolution became limiting. For the present, this absolute warming threshold may be higher, as a colder background climate state likely results in the positive effects of warming and CO₂ fertilization on productivity we observe for the PETM extending to lower latitudes. However, with carbon currently being released at a rate likely exceeding the PETM by around a magnitude, the speed of dispersal may become much more important at present for inducing an adaptive lag. In summary, we believe absolute warming caused the adaptive lag during the PETM, whereas at present, the rate of warming may be considerably more important, despite a potentially higher absolute warming threshold beyond which evolution becomes limiting.

Turner, S. K. Constraints on the onset duration of the Paleocene–Eocene Thermal Maximum. *Philosophical Transactions of the Royal Society A: Mathematical, Physical and Engineering Sciences* 376, 20170082 (2018).

B - implications for the interpretation of the $\delta^{13}\text{C}$ anomalies during the PETM

As the authors mention multiple times, $\delta^{13}\text{C}$ changes in marine carbonates are a key feature of the geologic record of the PETM. It has even been used to estimate the magnitude of carbon fluxes during the event, usually without considering terrestrial $\delta^{13}\text{C}$ fluxes (e.g. Gutjahr et al. 2017, Komar and Zeebe, 2017, Haynes et al. 2020). The authors already allude to it but I think it would be interesting and could increase the relevance and reach of this paper by adding a brief discussion of the implications of the simulated terrestrial carbon stock changes in the present study would affect these estimates, and to put the simulated carbon fluxes in relation to those previously estimated.

We have extended and clarified the discussion on how our modelled terrestrial carbon stock changes compare to proxy-based carbon budgets and flux estimates for the PETM (L365-376). Recent studies suggest that approximately 5800 Pg of carbon were released during the PETM, that isotopically light carbon likely contributed substantially to the total carbon release (in addition to volcanic carbon and methane), and that a total organic carbon sequestration of 2000-2500 Pg of organic carbon would have been necessary to explain the $\delta^{13}\text{C}$ records during the PETM recovery. Our simulations estimate a total carbon pool of 2000-2500 Pg of carbon in vegetation and soils and that 50% or more of this carbon could have been released during the PETM onset, followed by a lagged regrowth. Thus, according to our results, the terrestrial carbon dynamics may have contributed substantially to the global dynamics as derived from proxy records.

2) I think it would be a valuable addition to discuss how these results depend on the background state, e.g. how applicable they are to other climate events. The PETM starts from a much warmer background climate than today, with higher atmospheric CO_2 . How does this affect the potential of plants to adapt or migrate under warming compared to today? Does the CO_2 fertilization effect tail off when atmospheric CO_2 is above a certain threshold?

We have taken up the aspect of the background climate state in the comparison of the PETM and current climate change (L400-426). During the PETM, we show that increased temperatures and CO_2 fertilization may have shifted conditions in the highest latitudes towards more optimal conditions for photosynthesis and productivity. In a generally colder climate and a lower initial atmospheric CO_2 concentration, we expect these effects to extend further into lower latitudes. For a colder background climate state, we further expect that it requires more warming to have widespread regions where temperature and precipitation conditions are unsuited for primary productivity (e.g., temperatures exceeding critical thresholds for photosynthesis).

3) Does the vegetation model consider disturbances like fire, storm or pest damage, which might have increased due to the PETM warming? If not, how might including these processes alter the results? Do they represent a best-case scenario?

We thank the reviewer for highlighting this aspect. The current model does not consider such additional disturbances, which might have played a role during the PETM. We can expect that for a given rate of adaptation in the model, additional disturbances would lower the productivity and carbon sequestration and prolong the lag in effective climate regulation. We have added this aspect to the discussion (L339-340).

References

Gutjahr, M., Ridgwell, A., Sexton, P.F., Anagnostou, E., Pearson, P.N., Pälike, H., Norris, R.D., Thomas, E. and Foster, G.L., 2017. Very large release of mostly volcanic carbon during the Palaeocene–Eocene Thermal Maximum. *Nature*, 548(7669), pp.573-577.

Haynes, L.L. and Hönlisch, B., 2020. The seawater carbon inventory at the Paleocene–Eocene Thermal Maximum. *Proceedings of the National Academy of Sciences*, 117(39), pp.24088-24095.

Komar, N. and Zeebe, R.E., 2017. Redox-controlled carbon and phosphorus burial: A mechanism for enhanced organic carbon sequestration during the PETM. *Earth and Planetary Science Letters*, 479, pp.71-82.

Reviewer #3 (Remarks on code availability):

The code is very well documented and seems complete. I didn't test all scripts but the one I tested worked fine.

Reviewer #1 (Remarks to the Author):

I reviewed a previous version of this manuscript. The authors have addressed all of my comments in both their response and revision. I think this is an excellent study and worthy of publication in its present form.

We thank the reviewer for the helpful comments during the review process.

Reviewer #2 (Remarks to the Author):

Review of: "Loss of vegetation functions during the Paleocene-Eocene thermal maximum" by Rogger, J., Korasidis, V. A., Bowen, G. J., Shields, C. A., Gerya, T. V., and Pellissier, L., manuscript submitted to Nature Communications

I read carefully the revised version of the manuscript. The authors addressed all the comments I made in my first review and modified the main manuscript and the supplementary text accordingly. From my point of view, the points raised by the other reviewers were also quite fully addressed. As a result, the paper can be published in its current form, after corrections of a few typos/grammatical errors I found while reading:

Page 16, line 369: "a carbon cycle" instead of "a carbon carbon cycle"

Changed accordingly.

Page 17, line 411: "by around an order of magnitude" instead of "by around a magnitude"

Changed accordingly.

Page 20, line 462: "are used to describe" instead of "are used describe"

Changed accordingly.

Page 48, Legend of Supplementary Figure 6, A & B: the figures show LMA versus latitude, while the legend refers to SLA. All occurrences of SLA in the legend should be replaced by LMA. Also, legend for figure D is missing.

The legend of Supplementary Figure 6 has been revised accordingly, and a description of panel D has been added.

We thank the reviewer for the helpful comments during the review process.

Reviewer #3 (Remarks to the Author):

I thank the author team for their efforts in revising the manuscript. I re-read the revised manuscript a few times now and think that, while future modelling studies should take more of the complexity of the coupled atmosphere-land-ocean system into account, this current manuscript presents a well rounded study and will surely spark important debates. I look forward to seeing this manuscript published.

We thank the reviewer for the helpful comments during the review process.

Reviewer #3 (Remarks on code availability):

I could not access the source code via the provided link

We have checked the link, and all code files and data are accessible under:
<https://zenodo.org/records/17252973>.